# DIRECT OPTIMAL ACTION LEARNING

## ABSTRACT

Recent advancements in offline reinforcement learning leveraged two key innovations: policy extraction from behavior-regularized actor-critic (BRAC) objective and expressive policies, such as diffusion and flow models. However, backpropagation through iterative sampling chains is computationally tricky and often requires policy-specific solutions and careful hyperparameter tuning. We observe that the reparameterized policy gradient of the BRAC approximately trains the policy to replicate an 'optimal' action. Building on this insight, we introduce **Direct Optimal Action Learning (DOAL)**, an efficient, effective, and versatile framework for policy extraction from Q value functions. DOAL utilizes efficient behavior losses native to the policy's distribution (e.g., flow matching loss) to imitate an optimized action based on Q-values. Furthermore, we demonstrate that the traditional balancing factor between Q-loss and behavior-loss can be reinterpreted as a mechanism for selecting a trust region for the optimal action. The trust region reinterpretation yields a **Batch-Normalizing Optimizer**. This facilitates the hyperparameter search and makes it shareable across polices. Our DOAL framework can be easily integrated with any existing Q-value-based offline RL methods. We apply DOAL to Gaussian, Diffusion, and Flow policies. For Diffusion and Flow policies, our baseline models use the MaxQ action sampling, where the **number of samples** is tuned for each task. In particular, with regularized Q value estimation, flow policies achieved the best results. On 9 OGBench tasks, our baseline models outperformed the previous best models, and DOAL improves over strong baseline models while simplifying hyperparameter search. On 6 Adroit tasks from D4RL, improvement can be achieved when the Q value learning is regularized. The code is available through Anonymous Github.

## 1 INTRODUCTION

Offline reinforcement learning (RL) aims at efficiently and effectively extracting policy from experience beyond the simple imitation learning(Lange et al., 2012; Levine et al., 2020). For offline RL agents to perform better than a simple behavior cloning agent, value estimation and extracting information from the estimated value function is the key citep park2024is. Yet, due to the lack of interaction, agents cannot extrapolate too much to avoid the distribution shift. Hence, the success of offline RL depends on balancing the maximization of the estimated Q value and behavior cloning (Haarnoja et al., 2018; Wu et al., 2019; Kostrikov et al., 2022; Tarasov et al., 2023).

Meanwhile, as the scale and diversity of offline datasets continue to grow, there is an increasing need for policies capable of modeling highly multi-modal and complex action distributions. Diffusion and flow matching models(Sohl-Dickstein et al., 2015; Ho et al., 2020; Song et al., 2021b; Lipman et al., 2023; Lu et al., 2022; Sun et al., 2025) have recently been applied to Offline RL(Janner et al., 2022; Wang et al., 2023; Hansen-Estruch et al., 2023; Kang et al., 2023; Park et al., 2025c), and improved the modeling capacity of policy distribution.

However, efficient policy extraction requires a reparameterized policy gradient through the Q value function(Park et al., 2024), and it is non-trivial for distribution involving an iterative sampling procedural(Park et al., 2025c; Kang et al., 2023; Fujimoto & Gu, 2021; Tarasov et al., 2023). While many solutions have been proposed, they usually involve some computation overhead, require careful tuning of hyperparameter across environments, or are not as effective (Wang et al., 2023; Chen et al., 2023; Kang et al., 2023; Lu et al., 2023).

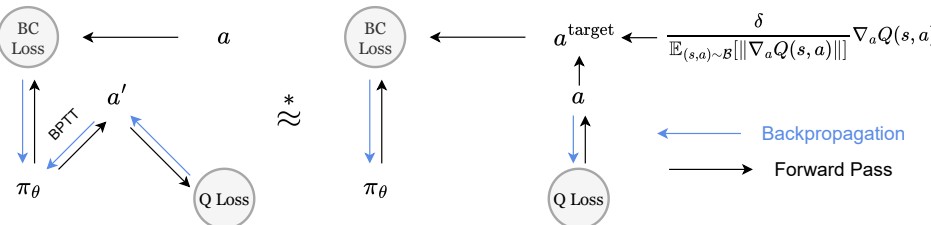

Figure 1: **DOAL Framework.** On the left, we have the end2end re-parameterized policy gradient from the Q and the behavior clone loss. On the right, we have our DOAL framework, where we extract an optimized $a^{\text{target}}$ from Q, and use policy dependent efficient behavior clone loss.

In this work, we present an efficient, effective, and versatile framework for policy extraction from Q-value functions. As shown in Figure 1, we observe that the reparameterized policy gradient w.r.t. *Behavior Regularized Actor Critic (BRAC)* objective can be replaced by simple gradient w.r.t. behavior clone loss of an optimized action. [1] In Proposition 1, we make a slightly more formal statement about their similarity and difference when the BC loss is the Mean Squared loss.

This observation has two consequences: **Direct Optimal Action Learning (DOAL)** instead of backpropagating end2end, we can optimize the action with q value and behaviour loss, then learn the optimized action with the efficient loss function that is native to the policy distribution (e.g., velocity matching loss); **Batch-Normalizing Optimizer** if learning from Q value function is approaching a better local solution, and Q value estimations are known to be not reliable, we should have a trust region $\delta$ for how much the optimized $a^{\text{target}}$ can shift away from data distribution. Meanwhile, it is also desirable that the shift distance be proportional to the gradient of the Q value w.r.t. action. This can be realized by normalizing the shift with a batch average of gradient $a^{\text{target}} = a + \frac{\delta}{\mathbb{E}_{(s,a)\sim\mathcal{B}}[\|\nabla_a Q(s,a)\|]}\nabla_a Q(s,a)$, where $\mathcal{B}$ is the data mini-batch.

To isolate the effects of value estimation from policy extraction, we use the *Implicit Q Learning (IQL)* (Kostrikov et al., 2022). The value estimation in IQL doesn't interact with the policy extraction, making it an ideal choice for our study. Still, the learned Q-value can be used. For Gaussian policy, *Advantage-Weighted Regression (AWR)* can use Q-value to weight actions during training (Kostrikov et al., 2022). For flow and diffusion models, we use *MaxQ Sampling* to select the action with the highest Q-value from a set of sampled candidates during testing. To obtain strong baseline models, we recognized that the **sample size for action candidates** is a critical, previously neglected hyperparameter (Park et al., 2025c). Varying this number allows us to manage the inherent trade-off between the overestimation risk in the Q-value (maximization bias) and the representativeness of the sample set (coverage). To obtain better performance, we further tested DOAL framework with Q-Learning and regularized Q-Learning.

Empirically, in all, we tested over three different Q-value Empirically, we tested over three different Q-value functions (IQL, Q-Learning, and Regularized Q-Learning) and three classes of policies (Gaussian, flow, and diffusion models). We performed experiments in 9 default tasks on OGBench and 6 Adroit tasks on D4RL datasets. Overall, the DOAL learned policies achieved improvement over strong baselines on OGBench. On Adroit tasks, we observed performance gain with regularized Q-learning. Both our baseline models and DOAL models suppress the previously best published work, FQL (Park et al., 2025c). Importantly, for all algorithms in the same task and same value function, the DOAL hyperparameters $\delta$ are shared, and DOAL costs one extra forward and backward call of the Q value net, compared to baselines. In all, the DOAL framework is an **efficient, effective** and **versatile** tool to extract polices from Q values.

## 2 PRELIMINARIES

We consider a Markov decision process (MDP) $\mathcal{M} = (\mathcal{S}, \mathcal{A}, r, p, \gamma, \rho_0)$(Sutton & Barto, 1998), where $\mathcal{S}$ is the state space, $\mathcal{A} = \mathbb{R}^d$ is a $d$-dimensional continuous action space, $r(s,a)$ is the reward

---

[1]Notice that we are not claiming the two objectives are equivalent, but rather the BRAC and DOAL objective both pushes the policy to produce higher valued action while being close to the action data point.

function, $p(s' \mid s, a)$ is the transition dynamics, $\gamma \in [0, 1)$ is the discount factor, and $\rho_0(s)$ is the initial state distribution. In offline RL, the agent learns from a fixed dataset $\mathcal{D} = \{(s_i, a_i, s'_i, r_i)\}_{i=1}^N$ consisting of individual transitions rather than complete trajectories. The objective is to learn a policy $\pi_\theta(a \mid s)$ that maximizes the expected discounted return $J(\pi_\theta) = \mathbb{E}_{\pi_\theta} [\sum_{t=0}^\infty \gamma^t r(s_t, a_t)]$, while avoiding distributional shift caused by querying actions outside the support of $\mathcal{D}$.

**Implicit Q-Learning.** IQL (Kostrikov et al., 2022) avoids querying out-of-sample actions through expectile regression. Unlike SARSA-style methods (Brandfonbrener et al., 2021) that learn using the next actions from the dataset, IQL learns the value function of the current policy.

The value function $V_\psi(s)$ and the Q-function $Q_\phi$ are learned via:

$$\mathcal{L}_V(\psi) = \mathbb{E}_{(s,a)\sim\mathcal{D}} \left[ L_2^\tau \left( Q_\phi(s, a) - V_\psi(s) \right) \right], \quad \mathcal{L}_Q(\phi) = \mathbb{E}_{(s,a,r,s')\sim\mathcal{D}} \left[ \left( r + \gamma V_\psi(s') - Q_\phi(s, a) \right)^2 \right]. \tag{1}$$

where $L_2^\tau(u) = |\tau - \mathbb{I}(u < 0)|u^2$. For policy extraction, standard IQL uses Advantage-Weighted Regression (AWR) (Peng et al., 2019):

$$\mathcal{L}_{\text{AWR}}(\theta) = \mathbb{E}_{(s,a)\sim\mathcal{D}} \left[ \exp \left( \beta(Q_\phi(s, a) - V_\psi(s)) \right) \log \pi_\theta(a|s) \right], \tag{2}$$

where $\beta$ controls the strength of the advantage weighting.

**Flow Matching.** Flow Matching (FM) establishes a deterministic path $(p_t)_{t\in[0,1]}$ that continuously transforms a simple source distribution $p_1$ into the target behavior distribution $p_0$, with each $p_t$ defined over $\mathbb{R}^d$ (Lipman et al., 2023; 2024). [2]

We adopt the most simple instantiation of FM, utilizing linear interpolation paths with uniform time sampling (Lipman et al., 2024; Park et al., 2025c). For an action $a_0 \sim \pi(a_0)$, we learn a time-dependent velocity field $v_\theta(a, t) : \mathbb{R}^d \times [0, 1] \to \mathbb{R}^d$ through the following regression loss:

$$\mathcal{L}_{\text{FM}}(\theta) = \mathbb{E}_{a_0\sim q(a_0), a_1\sim\mathcal{N}(0,I), t\sim\mathcal{U}[0,1]} \left[ \|v_\theta(a_t, t) - (a_0 - a_1)\|^2 \right], \quad a_t = (1 - t)a_0 + ta_1 \tag{3}$$

This formulation ensures training stability while admitting efficient sampling through explicit Euler discretization of the underlying flow ODE:

$$a_{t-\Delta t} = a_t + \Delta t \cdot v_\theta(a_t, t), \quad \Delta t = \frac{1}{N} \tag{4}$$

where $N$ represents the number of flow steps.

**TrigFlow.** We adapt the TrigFlow (Lu & Song, 2025) to train diffusion policy(Sohl-Dickstein et al., 2015; Ho et al., 2020; Song et al., 2021b), defining the forward diffusion process as follows: given $a_0 \sim p(a_0)$ and $z \sim \mathcal{N}(0, I)$, the noisy sample at time $t$ is given by

$$a_t = \cos(t)a_0 + \sin(t)z, \quad t \in [0, \frac{\pi}{2}] \tag{5}$$

with $a_{\pi/2} \sim \mathcal{N}(0, I)$. The model can be trained to minimize the prediction error:

$$\mathcal{L}_{\text{diffusion}}(\theta) = \mathbb{E}_{a_0\sim p(a_0), z\sim\mathcal{N}(0,I), t\sim\mathcal{U}[0,\pi/2]} \left[ \|f_\theta(a_t, t) - a_0\|_2^2 \right], \tag{6}$$

$$f_\theta(a_t, t) = \cos(t)a_t - \sin(t) \cdot F_\theta(a_t, t), \tag{7}$$

where $F_\theta$ is a learned network and $f_\theta$ naturally satisfies the boundary condition $f_\theta(a_0, 0) = a_0$. We implement a first-order DDIM(Song et al., 2021a) sampling scheme for efficient inference:

$$a_t = \cos(k - t) \cdot a_k - \sin(k - t) \cdot F_\theta(a_k, k), \tag{8}$$

where $k$ is the previous time step. In our paper, we divide the steps evenly into 10.

**Max Q Sampling.** A principled strategy orthogonal to behavior-regularized actor-critic frameworks is a resampling mechanism (Ghasemipour et al., 2021; Chen et al., 2023; Hansen-Estruch et al., 2023; Li et al., 2025). Instead of training the action policy with Q value, Max Q sampling leverages the estimated $Q_\phi$ in the inference time. Formally, given a proposal distribution $\pi_\theta(a|s)$ and a target criterion $Q_\phi(s, a)$, the Max Q sampling procedure selects the optimal action from a set of samples:

$$a = \arg\max_{a^{(1)},...,a^{(n_{\text{sample}})}} Q_\phi(s, a^{(i)}), \quad a^{(i)} \sim \pi_\theta(s) \tag{9}$$

---

[2]In the usual flow matching notation $p_1$ is the data distribution, but we want to make it consistent with diffusion model notations.

**Behavior-Regularized Actor-Critic (BRAC).** Behavior-regularized actor-critic methods form a family of effective offline RL approaches that combine value function learning with behavior regularization (Haarnoja et al., 2018; Wu et al., 2019; Tarasov et al., 2023). The critic loss follows equation 1. The actor loss, defined in our implementation, integrates policy improvement and behavior regularization:

$$\mathcal{L}_{\pi,\text{BCLoss}}^{\text{BRAC}}(\theta) = \mathbb{E}_{(s,a)\sim\mathcal{D}}\left[-Q_\phi(s,\pi_\theta(s)) + \alpha \cdot \text{BCLoss}(\pi_\theta(s),a)\right], \tag{10}$$

where $\alpha$ is a hyperparameter balancing policy improvement and behavior constraint and BCLoss is a behavior clone loss, e.g. $\|\pi_\theta(s) - a\|^2$ or the velocity matching loss in the previous section. While the BARC objective is almost necessary, it has been shown to be highly sensitive to $\alpha$ (An et al., 2021; Chen et al., 2024b; Fang et al., 2025; Gao et al., 2025), requiring extensive per-task tuning.

The problem in applying Equation 10 to a diffusion/flow-based policy is the computationally costly iterative sampling chain(Sohl-Dickstein et al., 2015; Ho et al., 2020; Song et al., 2021b). There exists many works to circumvent this issue, see Section 6 for more discussion.

**Regularized-Behavior-Regularized Actor-Critic (ReBRAC).**

$$\mathcal{L}(\phi) = \mathbb{E}_{(\mathbf{s},\mathbf{a},r,\mathbf{s}',\mathbf{a}')\sim\mathcal{D}}\left[\|Q_\phi(\mathbf{s},\mathbf{a}) - \left(r + \gamma\left(Q_{\phi'}(\mathbf{s}',\pi_\theta(\mathbf{s}')) - \alpha_{\text{critic}} \cdot (\mathbf{a}' - \pi_\theta(\mathbf{s}'))^2\right)\right)\|_2^2\right], \tag{11}$$

where $Q_{\phi'}$ is target critic network and the $\alpha_{\text{critic}}$ is the critic bc coefficient. This regularized Q-value target reduces the overestimation issue in Q-learning (Tarasov et al., 2023).

For notational simplicity, we drop the mention of noise in the main text, as that can be easily integrated, or we consider a joint state $s' = (s, z)$ for where noise is needed.

# 3 DIRECT OPTIMAL ACTION LEARNING

In deep learning, there is a doctrine that all we need to train neural networks is a differentiable loss w.r.t. the parameters (Rumelhart et al., 1995). In the context of learning policy with iterative sampling, the computational cost becomes too high. In fact, it is not entirely clear what the learned policy represents aside from it balances behavior regularization and expected return maximization. We present a intuitive motivation and a formal motivation to derive our Direct Optimal Action Learning framework, and show how it can naturally yields more interpretable and robust hyperparameter choice.

## 3.1 THE OPTIMAL ACTION

**Intuitive Perspective.** An actor policy should learn to produce actions that have high estimated Q-values. When learning from a static dataset, this search for high-value actions should naturally be centered around the actions already present in the data. This leads to a straightforward idea of direct optimal action learning (DOAL): instead of indirectly learning a policy that optimizes a complex objective, we can directly learn the optimal action itself. In related works 6, we discuss other neural network learning methods that also follow non-conventional target matching objectives.

**Formal perspective.** The DOAL idea emerges from a re-interpretation of the policy gradient in standard behavior-regularized actor-critic methods:

**Proposition 1** (BRAC Objective Pursue BRAC Target). *Let $\pi_\theta(s)$ be a deterministic differentiable policy and $Q_\phi(s, a)$ be a differentiable action-value function. Consider the behavior-regularized objective:*

$$\mathcal{L}_Q(\theta) \triangleq \mathbb{E}_{(s,a)\sim\mathcal{D}}\left[Q_\phi(s,\pi_\theta(s)) - \alpha\|\pi_\theta(s) - a\|_2^2\right] \tag{12}$$

*where $\alpha > 0$ is a regularization coefficient. The gradient of this objective with respect to $\theta$ is equivalent to the gradient of a simpler squared-error objective $\mathcal{L}_{target}(\theta)$:*

$$\nabla_\theta \mathcal{L}_Q(\theta) = \nabla_\theta \mathcal{L}_{\text{brac\_target}}(\theta) \triangleq \mathbb{E}_{(s,a)\sim\mathcal{D}}\left[\nabla_\theta\left(-\alpha\left\|\pi_\theta(s) - a^{\text{brac\_target}}\right\|_2^2\right)\right] \tag{13}$$

$$a^{\text{brac\_target}} = a + \frac{1}{2\alpha}\nabla_{a'}Q_\phi(s,a')\big|_{a'=\pi_\theta(s)} \tag{14}$$

The proof is an application of the chain rule (see Appendix B). This gradient equivalence reveals that the BRAC objective implicitly minimizes the distance between the policy's output $\pi_\theta(s)$ and a target action $a^{\text{brac-target}}$. Under such an interpretation, a conceptual inconsistency exists. The target $a^{\text{brac-target}}$ is constructed by taking a gradient ascent step from the data action $a$, but the gradient $\nabla_{a'}Q_\phi$ is evaluated at the policy's output $\pi_\theta(s)$. This requires action sampling at training time and creates a mismatch between the point of expansion ($a$) and the point of Q value gradient evaluation ($\pi_\theta(s)$). DOAL performs the gradient ascent using the gradient evaluated at the data action $a$, which would define a $a^{\text{target}}$ for each data point $(s, a)$ without sampling. Importantly, Proposition 1 shows BRAC objective and the DOAL objective are *similar but different*. DOAL is a reasonable objective for offline RL in its' own right.

By defining the target action directly from the data and Q value, DOAL decouple the target computation from the policy being trained. The major practical benefit is that we no longer need to sample an action during training. Consequently, we can leverage more powerful generative modeling techniques to learn the action distribution, such as training flow-based models with flow matching losses or diffusion models with their diverse loss functions.

## 3.2 BATCH-NORMALIZING OPTIMIZER

A challenge with BRAC-style methods is the sensitivity of the regularization coefficient $\alpha$, which often requires careful tuning across several orders of magnitude (Park et al., 2025c; Kumar et al., 2020). Our re-interpretation of the objective as learning a target action raises a critical question: what is the appropriate magnitude for the update from the data action $a$ to the target action $a^{\text{target}}$?

In offline reinforcement learning, the learned Q-function is merely an estimator, and it is crucial to remain conservative to avoid distribution shift to out-of-support actions where the Q-function is unreliable (Kostrikov et al., 2022; Kumar et al., 2020; Tarasov et al., 2023). Therefore, instead of an arbitrary step size dictated by $\alpha$, we should define a **statistical trust region** for our action optimization. We can achieve this by setting a fixed expected magnitude for the update vector $g(s, a) = a^{\text{target}} - a$. Specifically, we desire two conditions for the update vector $g(s, a)$:

1. The update should be in the direction of the Q-function's gradient at the data action: $g(s, a) = C \cdot \nabla_a Q_\phi(s, a)$ where $C > 0$.
2. The expected squared magnitude of the update over the dataset should be a constant, which we denote as $\delta$: $\mathbb{E}_{(s,a)\sim\mathcal{D}}[\|g(s, a)\|_2] = \delta$.

These two conditions uniquely determine the update vector, as shown in the following proposition.

**Proposition 2** (Batch Normalized Update). *To satisfy the conditions $g(s, a) = C \cdot \nabla_a Q_\phi(s, a)$ and $\mathbb{E}_{(s,a)\sim\mathcal{D}}[\|g(s, a)\|_2] = \delta$, where $\delta, C > 0$, the only action update vector $g(s, a)$ is :*

$$g(s, a) = \frac{\delta}{\mathbb{E}_{(s',a')\sim\mathcal{D}}\left[\|\nabla_{a'}Q_\phi(s', a')\|_2\right]} \cdot \nabla_a Q_\phi(s, a) \tag{15}$$

The proof is straightforward calculation (see Appendix C). This formulation replaces the obscure hyperparameter $\alpha$ with an interpretable one, $\delta$, which directly controls the expected squared magnitude of the action update. In practice, we use the batch statistics as estimator, so we have $\mathbb{E}_{(s',a')\sim\mathcal{B}}\left[\|\nabla_{a'}Q_\phi(s', a')\|_2^2\right]$, where $\mathcal{B}$ is the current mini-batch. This "batch-normalization" of the action gradients provides a more stable and robust training target, alleviating the need for extensive hyperparameter sweeps. *We are not claiming that this batch normalized scheme can find better $a^{\text{target}}$ than not using batch-normalized gradient.* In fact, if the gradient statistics is stable, you can always get the same result by having $g(s, a) = C \cdot \nabla_a Q_\phi(s, a)$ where $C := \frac{\delta}{\mathbb{E}_{(s',a')\sim\mathcal{D}}[\|\nabla_{a'}Q_\phi(s',a')\|_2]}$. We will empirically see that optimal $\delta$ varies less than $\delta'$.

## 3.3 THE DOAL OBJECTIVES

DOAL can work with any value function learning. We only alter the actor loss :

$$\mathcal{L}_{\text{DOAL}}(\theta) = \alpha \cdot \mathbb{E}_{(s,a)\sim\mathcal{D}}\text{BCLoss}(\pi_\theta(s), a^{\text{target}}) \tag{16}$$

$$a^{\text{target}} := a + \frac{\delta}{\mathbb{E}_{(s',a')\sim\mathcal{B}}\left[\|\nabla_{a'}Q_\phi(s', a')\|_2\right]} \cdot \nabla_a Q_\phi(s, a) \tag{17}$$

where BCLoss can be any distribution matching losses. In terms of computational overhead, as the gradient $\nabla_a Q_\phi(s, a)$ will be called explicit or implicitly at least once to extract first order structure information from $Q_\phi$. We still keep the $\alpha$ parameter from (Park et al., 2025c) for all experiments for consistency. In ablation study in Appendix F, we find setting it to 1 is fine.

## 4 MaxQ Sampling needs Balancing

As our DOAL framework learn from optimized action based on $\nabla_a Q(s, a)$, we consider models without learning from $\nabla_a Q(s, a)$ as our baseline models. In such case, $Q(s, a)$ can still be used for weighting the behavior clone loss and performing actions selection, as we discussed Section 2. To have a solid baseline for comparison, we argue that $n_{\text{sample}}$ a crucial hyper-parameter.

When taking $n_{\text{sample}} = 1$, we clearly recover the actor distribution and completely lost access to information from $Q_\phi$. On the other extreme, Some earlier work suggested the bigger $n_{\text{sample}}$, the better (Ghasemipour et al., 2021). However, while $a^*$ maximize the Q value estimator $Q_\phi$ as $n_{\text{sample}} \to +\infty$, there exists maximization bias such that the maximum Q value will be overestimated due to noise in the estimator, and $a^*$ with large positive random fluctuation will be selected.

**Proposition 3** (Informal). *Consider countably many actions $a_1, a_2, \ldots$. For each $i$, the Q-estimator is independent Gaussian: $\widehat{Q}(a_i) \sim \mathcal{N}(\mu_i, \sigma_i^2)$, with bounded means $\sup_i \mu_i \leq U < \infty$ and non-trivial noise $\sigma_i \geq c > 0$. Draw $n$ i.i.d. actions from any policy with dense support and pick the one with the largest observed $\widehat{Q}$. As $n \to \infty$:*

- *(i) the selected $\widehat{Q}$ value diverges to $+\infty$ (driven by extreme positive noise);*

- *(ii) the probability of picking any true-mean maximizer tends to $0$.*

**Proof Intuition.** The maximum of $m$ i.i.d. Gaussian draws grows like $\mu_i + \sigma_i\sqrt{2\log m}$. With infinitely many actions having $\sigma_i > 0$, actions with an extremely large positive fluctuation will dominate the max, regardless of the bounded means. Thus, making $n_{\text{sample}}$ large pushes selection toward noise outliers rather than actions with the highest $\mu_i$, even with unbiased estimator. $\qquad \square$

The above analysis shows that increasing $n_{\text{sample}}$ can exacerbate the maximization bias: as $n_{\text{sample}} \to \infty$, $\max$-selection over noisy Q-estimates systematically prefers actions with large positive noise realizations, independent of the learned $Q$. While it is hard to precisely characterize the resulting distribution from MaxQ sampling, the more samples we have the more data coverage we have. In the case where action policy is value agnostic, having multiple samples is necessary to ensure mode coverage so that Q function can provide a selection. Yet, if we sample too much, the stochasticity in $Q_\phi$ dominates and we might get "good" actions due to overestimation rather than being reliable. Pragmatically, $n_{\text{sample}}$ balances the conflict between the in-distribution behavior and the noisy Q value estimator in the inference time.

## 5 Experiments & Discussion

In this section, we examine our DOAL framework. Our baseline models benefit from $Q_\phi(s, a)$ either through advantage weighted regression or max-Q sampling without accessing to $\nabla Q_\phi(s, a)$. We analyse the time complexity of DOAL, and discuss the trust region $\delta$ selection. Other ablation studies are presented in Appendix F.

**Benchmarks** We conduct a comprehensive evaluation of our method on two challenging offline RL benchmarks: 1) The 6 Adroit/D4RL tasks with expert, human, and cloned dataset qualities. 2) 9 default tasks from the more diverse and demanding OGBench suite. We omit some tasks, as no current algorithms can work well (Park et al., 2025c). Following Park et al. (2025c), we train on OGbench for 1m steps, and take the average performance on 800k, 900k and 1m steps. We train on D4RL tasks for 500k steps and test on last step.

**Models** We test our DOAL framework under implicit Q-learning, Q-learning and regularized Q-learning. Our policy models include simple Gaussian policy, Multi-step Flow models and TrigFlow models. Except the number of MaxQ sampling candidates, we follow the hyperparameters from FQL (Park et al., 2025c) whenever possible, this includes $\tau$ in IQL and $\alpha$ for scaling BC loss.

Table 1: **IQL-based offline RL results.** Full results on the OGBench and D4RL tasks with IQL value function. They are the default single-task in each environment. The results are averaged over 8 seeds on D4RL and OGBench. The IQL(tanh)*, IQL(Gauss) on OGBench, IFQL* results are collected from Flow Q learning (Park et al., 2025c; Tarasov et al., 2022).

| Task | Simple Policies | | | Flow Policies | | | Diffusion Policies | | |
|---|---|---|---|---|---|---|---|---|---|
| | IQL(tanh)* | IQL(Gauss) | DIQL | IFQL* | IFQL | DIFQL | TrigFlow | DTrigFlow | ETrigFlow |
| antmaze-large-navigate | - | 48 ±9 | 63 ±10 | 24 ±17 | 48 ±24 | 67 ±6 | 72 ±6 | 63 ±24 | 63 ±21 |
| humanoidmaze-medium-navigate | - | 32 ±7 | 55 ±8 | 69 ±19 | 68 ±3 | 68 ±4 | 64 ±4 | 67 ±4 | 63 ±4 |
| humanoidmaze-large-navigate | - | 3 ±1 | 10 ±3 | 6 ±2 | 6 ±3 | 8 ±3 | 7 ±2 | 8 ±4 | 6 ±3 |
| antsoccer-arena-navigate | - | 3 ±2 | 13 ±3 | 16 ±9 | 40 ±5 | 40 ±6 | 40 ±8 | 41 ±4 | 41 ±9 |
| cube-single-play | - | 85 ±8 | 80 ±4 | 73 ±3 | 88 ±4 | 90 ±3 | 86 ±4 | 88 ±2 | 88 ±4 |
| cube-double-play | - | 1 ±1 | 3 ±2 | 9 ±5 | 11 ±3 | 21 ±4 | 16 ±4 | 22 ±3 | 16 ±3 |
| scene-play | - | 12 ±3 | 37 ±10 | 0 ±0 | 40 ±23 | 40 ±23 | 43 ±16 | 46 ±15 | 50 ±12 |
| puzzle-3x3-play | - | 2 ±1 | 5 ±1 | 0 ±0 | 5 ±1 | 5 ±2 | 7 ±2 | 7 ±3 | 8 ±2 |
| puzzle-4x4-play | - | 5 ±2 | 10 ±2 | 21 ±11 | 23 ±7 | 21 ±5 | 26 ±5 | 27 ±6 | 26 ±4 |
| Total | - | 191 | 276 | 218 | 329 | 359 | 361 | 368 | 359 |
| pen-human-v1 | 78 | 54 ±8 | 43 ±8 | 71 ±12 | 81 ±8 | 68 ±8 | 71 ±11 | 69 ±13 | 72 ±12 |
| pen-cloned-v1 | 83 | 66 ±7 | 56 ±9 | 80 ±11 | 73 ±7 | 74 ±7 | 65 ±7 | 67 ±8 | 67 ±9 |
| pen-expert-v1 | 128 | 131 ±8 | 132 ±4 | 139 ±5 | 134 ±4 | 138 ±4 | 135 ±8 | 133 ±7 | 134 ±8 |
| door-expert-v1 | 107 | 104 ±2 | 104 ±2 | 104 ±2 | 104 ±1 | 104 ±1 | 104 ±1 | 104 ±1 | 104 ±1 |
| hammer-expert-v1 | 129 | 68 ±47 | 76 ±46 | 117 ±9 | 96 ±8 | 98 ±12 | 103 ±8 | 98 ±11 | 100 ±10 |
| relocate-expert-v1 | 106 | 97 ±10 | 101 ±5 | 104 ±3 | 104 ±3 | 102 ±8 | 106 ±2 | 107 ±2 | 106 ±2 |
| Total | 631 | 520 | 518 | 615 | 592 | 584 | 584 | 577 | 583 |

For IQL value function, we have three representative and strong baseline algorithms with our method: IQL, with simple Gaussian policy learned from AWR; IFQL, a flow policy with MaxQ sampling; TrigFlow, a trigflow diffusion policy with MaxQ sampling. For each one of them, we have the DOAL versions DIQL, DIFQL, and DTrigflow. Furthermore, we have efficient trigflow (ETrigflow (Lu & Song, 2025; Kang et al., 2023),see Appendix D for details) that uses the BRAC objective but uses one-step sampling from corrupted $a_t$ and still use IQL for value learning.

For the Q-learning value function, our baseline model MFQL replace the IQL value learning with Q-learning. Also, we have the DOAL version DMFQL. Furthermore, we add the ReBRAC objective to regularize the Q learning target, and get MFReBRAC and DMFReBRAC (see Appendix D for details). We also report the results of $\text{MFQL}_{\text{bptt}}$ that actually running BPTT for actor learning.

## 5.1 MAIN RESULTS

Table 1 shows we have very strong baseline models of IFQL and Trigflow. In particular, our $\mathbf{n}_{\text{sample}}$ tuned IFQL improved over the original IQL* significantly on OGBench. On the OGBench, **on aggregation, our DOAL models performed better than their baselines.** Up on closer examination, we find that those are due to one or two tasks that has significant gains'that we high-lighted. Otherwise, their performance is very similar. On antmaze-large, both DTrigFlow and ETrigFlow dropped their performance. We find that there are two seeds that have very low performance (hence the large std). For training curves see Appendix E.

On the D4RL, we re-run IQL from FQL paper. It appears that there is no performance gain from either DOAL model or even ETrigflow. This might be due to the unreliability of IQL learned function gradient. It should be noted that the DOAL models subsume their baselines, as one can set $\delta = 0$ to **recover** the baseline or choose extremely small $\delta$. However, even if finer search for $\delta$ can yield higher performance, one can not rule out the selection bias. A proper way to address the inability to extract information from Q value is to investigate (Regularized) Q-Learning based functions.

In Table 2, excluding MFQL with BPTT, [3] all our models outperform FQL. In fact, in Table 2, MFQL outperforms FQL. This shows that effectiveness of MaxQ sampling. Furthermore, DMFQL outperforms MFQL on OGBench but not on D4RL. This again indicates that the effectiveness of DOAL might depend on the task or quality of Q value function. The fact, DMFReBRAC outperforms MFReBRAC indicates well regularized Q function can make DOAL work better. We present MFQL with BPTT to show that while being less stable, it is not always weaker. However, the computational complexity of BPTT is indeed much higher as we will discuss.

In Figure 4, we have the relations between MFQL,DMFQL, MFReBRAC and DMFReBRAC. We should notice that the DOAL version can achieve the same result as their baselines by letting $\delta = 0$, but we do not include such choice to explicitly show that first order gradient-based policy extraction might not always work.

---

[3]It might be possible that with better tuning on learning rates, BPTT training can achieve better performance. Nonetheless, it shows BPTT is fragile.

Table 2: **Q-Learning based offline RL results.** The results are averaged over 8 seeds . The ReBRAC*, FQL* results are collected from Flow Q learning (Park et al., 2025c).

| Task | Simple Policies | | Flow Policies | | | | | |
|------|-----------------|--------------|----------|----------|----------|----------------|----------|-----------|
| | ReBRAC(tanh)* | ReBRAC(Gauss) | FQL* | MFQL | DMFQL | MFQL$_{bptt}$ | MFReBRAC | DMFReBRAC |
| antmaze-large-navigate | 91 ±10 | - | 80 ±8 | 62 ±11 | 72 ±8 | 64 ±13 | 65 ±13 | 83 ±7 |
| humanoidmaze-medium-navigate | 16 ±9 | - | 19 ±12 | 49 ±9 | 44 ±13 | 54 ±9 | 53 ±14 | 52 ±7 |
| humanoidmaze-large-navigate | 2 ±1 | - | 7 ±6 | 8 ±3 | 7 ±3 | 6 ±2 | 9 ±4 | 8 ±2 |
| antsoccer-arena-navigate | 0 ±0 | - | 39 ±6 | 43 ±6 | 37 ±5 | 45 ±5 | 45 ±5 | 41 ±6 |
| cube-single-play | 92 ±4 | - | 97 ±2 | 95 ±1 | 98 ±1 | 62 ±37 | 91 ±5 | 99 ±1 |
| cube-double-play | 7 ±3 | - | 36 ±6 | 72 ±4 | 75 ±6 | 72 ±3 | 74 ±4 | 75 ±3 |
| scene-play | 50 ±13 | - | 76 ±9 | 57 ±20 | 90 ±10 | 68 ±15 | 57 ±12 | 92 ±6 |
| puzzle-3x3-play | 2 ±1 | - | 16 ±5 | 7 ±3 | 6 ±2 | 1 ±1 | 7 ±2 | 5 ±2 |
| puzzle-4x4-play | 10 ±3 | - | 11 ±3 | 24 ±3 | 14 ±4 | 0 ±0 | 25 ±5 | 12 ±3 |
| Total | 297 | - | 381 | 418 | 443 | 372 | 425 | 466 |
| pen-human-v1 | 103 | 55 ±9 | 53 ±6 | 75 ±9 | 72 ±8 | - | 64 ±9 | 74 ±8 |
| pen-cloned-v1 | 92 | 72 ±15 | 74 ±11 | 75 ±9 | 80 ±5 | - | 71 ±12 | 75 ±10 |
| pen-expert-v1 | 152 | 143 ±6 | 142 ±6 | 138 ±4 | 130 ±8 | - | 140 ±9 | 143 ±4 |
| door-expert-v1 | 106 | 105 ±1 | 104 ±1 | 104 ±2 | 104 ±1 | - | 105 ±8 | 105 ±1 |
| hammer-expert-v1 | 134 | 131 ±1 | 125 ±3 | 126 ±3 | 124 ±5 | - | 126 ±3 | 126 ±1 |
| relocate-expert-v1 | 108 | 107 ±1 | 107 ±1 | 106 ±4 | 104 ±4 | - | 106 ±1 | 107 ±2 |
| Total | 706 | 614 | 605 | 623 | 614 | - | 614 | 630 |

**Importance of Regularization.** On D4RL tasks in Table 2, we observe that only regularized Q function can boost the DOAL model performance. This strongly suggests that DOAL or maybe other gradient based policy extraction methods need the Q function to be reliable.

**Importance of Tanh.** Yet, our models still large behind ReBRAC that use a simple policy. We identified another difference that is the ReBRAC used tanh nonlinearity for producing actions between [-1,1]. This might introduce useful inductive bias. Indeed, by removing it, the performance of ReBRAC dropped. Studying how to add this nonlinear transformation after flow models is an interesting research question for future work.

## 5.2 TIME COMPLEXITY

| | FQL | IFQL | DIFQL | MFQL | MFQL-BPTT | DMFQL |
|--|-----|------|-------|------|-----------|-------|
| Forward Policy Call | 13 | 1 | 1 | 11 | 21 | 11 |
| Forward Value Call | 3 | 4 | 5 | 3 | 4 | 4 |
| Backward Policy Call | 2 | 1 | 1 | 1 | 10 | 1 |
| Backward Value Call | 2 | 2 | 3 | 1 | 2 | 2 |
| Total Calls | 20 | 8 | 10 | 16 | 37 | 18 |
| Actual Time in Minutes | 37 | 29 | 31 | 35 | 61 | 37 |

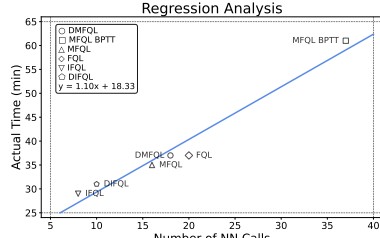

For MFQL, training policy net only requires 1 forward policy call. Train Q net requires sampling, therefore 10 forward policy calls. MaxQ sampling runs many samples in parallel, so as long as it fits to memory, it has no impact. MFQL takes 3 Value Q for Q learning and MaxQ sampling (yes, it can reduce to 2). DMFQL only adds one forward and one backward call to get $a^{target}$ compared to MFQL.

Figure 2: **Runtime and Computational Complexity .** On the left side, a table presents the training time number of function calls and actual runtime. On the right side, a regression line models the actual running time by the number of function calls. The actual run time is on the antmaze-large task with a single A800 GPU.

Offline RL algorithms train neural networks for value functions and policy distribution. In our work, 4-layer MLP are used for all neural modules. In Figure 2, we count the number of forward and backward calls through value and policy networks and present their total calls during the training phase. As forward and backward calls have similar computation cost and policy and value network share similar architectures, the sum of total calls can be used to predict the actual run time. Indeed, an affine relation is found. The relation is not linear, because there are overheads such as data loading and testing. One might notice that FQL's actual run time is faster than predicted, this is because FQL uses a one-step policy during testing. We did not include DMFReBRAC as it is the same as DMFQL. As for the memory usage, it is bottleneck-ed by the backward policy calls that requires storing intermediate computing states. Therefore, only BPTT version consume more memory.

## 5.3 CHOOSE STATISTICAL TRUST REGION $\delta$

In OGBench (Park et al., 2025a), actions are bounded in [-1,1] box of varying dimensionality. The statistical trust region should be related to how reliable are the Q value estimation at data point and how well can neural network generalize. As we are using IQL for value estimation

and the same value net in all our experiments, the only thing varying is the datasets. For OG-Bench experiments, we choose $\delta$ from $(0.03, 0.1, 0.3)$, and for D4RL experiments, we choose from $(0.0003, 0.001, 0.003)$ as we see from $\alpha$ in the FQL paper (Park et al., 2025c) tends to be extremely large. See Appendix G for the exact hyperparameters.

| Envs | puzzle-4x4 | cube-single | scene-play | antmaze-large-navigate |
|---|---|---|---|---|
| $\mathbb{E}_{(s',a')\sim\mathcal{B}}\left[\|\nabla_{a'}Q_\phi(s',a')\|_2\right]$ | 43.78 | 5.85 | 15.58 | 0.55 |
| $\alpha$ | 1000 | 300 | 300 | 10 |
| $\delta$ | 0.03 | 0.03 | 0.1 | 0.1 |
| $\frac{\delta}{\mathbb{E}_{(s',a')\sim\mathcal{B}}[\|\nabla_{a'}Q_\phi(s',a')\|_2]}$ | 6.9e-4 | 5.1e-3 | 1.9e-3 | 1.8e-2 |

Table 3: Different Environments with the Observed Mean $\|\nabla_a Q(s,a)\|$, $\alpha$, and $\delta$.

In Table 3, we present a few representative environments and their average $\|\nabla_a Q(s,a)\|$, optimal $\alpha$ (Park et al., 2025c) and $\delta$ we selected. As you can see the larger the gradient, the larger the selected $\alpha$. Based on our Proposition 1, we would be expecting $\alpha$ be inverse proportional to $\frac{\delta}{\mathbb{E}_{(s',a')\sim\mathcal{B}}[\|\nabla_{a'}Q_\phi(s',a')\|_2]}$. Indeed, this is what we observe. While $\alpha$ ranges across two orders of magnitude, our hyperparameter $\delta$ is relatively stable and easier to search for.

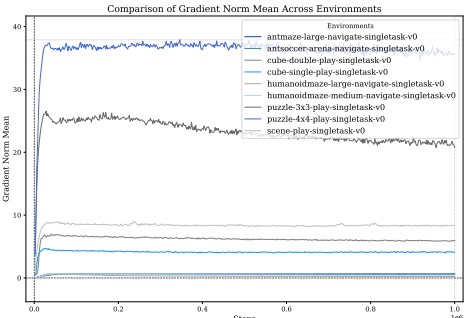

Figure 3: Mean Batch Normalized Gradient Norm of DMFQL across OGbench

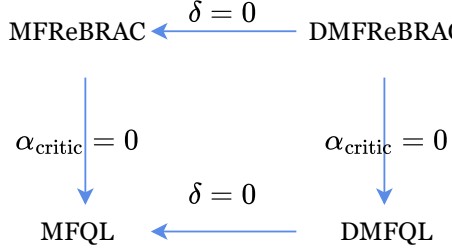

Figure 4: The relationship between MFQL, DM-FQL, MFReBRAC and DMFReBRAC.

In Figure 3, we can see the batch gradient normal is quite stable during training. This implies that $\frac{\delta}{\mathbb{E}_{(s',a')\sim\mathcal{B}}[\|\nabla_{a'}Q_\phi(s',a')\|_2]}$ is roughly a constant for each task, therefore, one can equivalently treating the direct gradient scaling factor as a hyperparameter and avoid the batch-normalization. The performance would be equivalent. However, the range of this linear scaling hyperparameter would be much wider. In Table 3, it ranges across two orders of magnitude just like $\alpha$.

## 6 RELATED WORK

### 6.1 REGULARIZED VALUE ESTIMATION

**In-Sample Optimization** (e.g., IQL(Kostrikov et al., 2022); X-QL(Garg et al., 2023); SQL and EQL(Xu et al., 2023) ) addresses offline RL by performing in-sample value iteration, avoiding queries to out-of-distribution (OOD) actions and directly approximating the optimal value function; **Conservative Methods** (e.g., CQL (Kumar et al., 2020); EDAC(An et al., 2021); AWAC(Nair et al., 2021); BCQ(Fujimoto et al., 2018); ReBRAC(Tarasov et al., 2023)) proposed methods that penalizing out-of-distribution actions to prevent overestimation of Q values.

### 6.2 POLICY EXTRACTION FOR DIFFUSION MODELS

**Accelerated Sampling Techniques.** A prominent research direction focuses on accelerating the sampling process of pre-trained generative policies. EDP (Kang et al., 2023) reformulates the reverse denoising process to estimate the target action $a_0$ in a single step. While these approaches

achieve notable improvements in sampling efficiency, they remain inherently heuristic. Meanwhile, FQL (Park et al., 2025c) circumvents the sampling via a student network with one-step sampling.

**Value Guidance Methods.** A substantial body of work in offline RL leverages diffusion models to approximate the behavior policy underlying the dataset. One prevalent strategy involves using the gradient of Q-value functions to guide the action generation process: QGPO (Wang et al., 2023), SFBC (Chen et al., 2023), EDA (Chen et al., 2024a), QVPO (Ding et al., 2024), and CFGRL(Frans et al., 2025). An alternative, more straightforward approach modulates the policy by re-weighting transition samples based on their estimated values (Peng et al., 2019; Frans et al., 2025).

**MaxQ Sampling.** MaxQ sampling has been introduced in early work with diffusion models. However, they mostly set $n_{\text{sample}}$ to be a large number and thought the trade-off is between computation budget vs. quality (Wang et al., 2023; Kang et al., 2023; Park et al., 2025c). Some other work used weighted resampling instead of MaxQ sampling(Hansen-Estruch et al., 2023), where proper weighting can alleviate the overestimation bias. As for MaxQ sampling, Ghasemipour et al. (2021) suggested larger $n_{\text{sample}}$ is better. A very recent/concurrent work on horizon reduction (Li et al., 2025) tuned the $n_{\text{sample}}$, arguing that for large $n_{\text{sample}}$, the sampled distribution might deviate from policy distribution too much. Yet, they did not discuss the over-estimation bias of the Q value.

## 6.3 POLICY DISTILLATION WITH REGULARIZED ACTOR-CRITIC METHODS.

Regularized value function learning has been shown to be effective (An et al., 2021; Hansen-Estruch et al., 2023). Contemporary methods like DAC (Fang et al., 2025) and BDPO (Gao et al., 2025) further integrate diffusion models with the regularized actor-critic framework. In particular, a concurrent work, FAC (Anonymous, 2025), builds on FQL and add carefully designed regularization on the Q value function, achieved very strong performance.

## 6.4 LONG HORIZON PROBLEMS

For challenging tasks that most algorithm cannot address, like antmaze-giant-navigate, (Park et al., 2023; 2025b; Li et al., 2025) looked at the horizon reduction issue. While this is orthogonal to our problem, such methods might be necessary for challenging environments.

## 6.5 NON END2END OPTIMIZATION

Similar to our DOAL at a conceptual level, there are several alternatives to end2end backpropagation that have been explored for neural network training. **Target-Propagation.** (Lee et al., 2014; Meulemans et al., 2020) propose to compute targets rather than gradients, at each layer. **Nested-Learning.** (Behrouz et al., 2025) trains neural networks with nested component-wise associative memory learning problems. It is not hard to see that their formulation can be converted to target matching like DOAL. **Nonparametric Learning.** (Wang et al., 2022) provably learns a two-layer network by matching the first layer output with the ideal second layer input.

## 7 CONCLUSION

In this work, we present Direct Optimal Action Learning, a framework that enables efficient and effective learning from $\nabla_a Q(s, a)$ for any policy distribution with different Q value functions. We provide strong baselines by re-exaiming the importance of $n_{\text{sample}}$ in MaxQ sampling, then we are able to show our models improved over baseline for OGBench tasks for various policies and value functions. On Adroit tasks, improvements are observed when we use ReBrac objective. Our experiment set up is mostly aiming at controlled study. In the future, better uncertainty aware Q estimation should be explored, as it might further improve the statistical trust region identification. Another important direction for diffusion/flow model training is to leveraging the squeezing layer such as tanh transformation.

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

## A    THE USE OF LARGE LANGUAGE MODELS

Large language models are used for polishing the texts, including equations and literature reviews.

## B    PROOF OF PROPOSITION 1

*Proof.* The proof follows from applying the chain rule. First, we compute the gradient of the regularized Q-objective in equation 12:

$$\nabla_\theta J_Q(\theta) = \nabla_\theta Q(s, \pi_\theta(s)) - \nabla_\theta \left( \alpha \|\pi_\theta(s) - a\|_2^2 \right)$$

$$= \left( \nabla_a Q(s, a')\big|_{a'=\pi_\theta(s)} \right)^\top \nabla_\theta \pi_\theta(s) - 2\alpha(\pi_\theta(s) - a)^\top \nabla_\theta \pi_\theta(s)$$

$$= \left[ \nabla_a Q(s, a')\big|_{a'=\pi_\theta(s)} - 2\alpha(\pi_\theta(s) - a) \right]^\top \nabla_\theta \pi_\theta(s)$$

Next, let the target action be $a^* \triangleq a + \frac{1}{2\alpha} \nabla_a Q(s, a')\big|_{a'=\pi_\theta(s)}$. We compute the gradient of the target-matching objective $J_{\text{target}}(\theta) = -\alpha \|\pi_\theta(s) - a^*\|_2^2$:

$$\nabla_\theta J_{\text{target}}(\theta) = -\nabla_\theta \left( \alpha \|\pi_\theta(s) - a^*\|_2^2 \right)$$

$$= -2\alpha(\pi_\theta(s) - a^*)^\top \nabla_\theta \pi_\theta(s)$$

$$= -2\alpha \left( \pi_\theta(s) - \left( a + \frac{1}{2\alpha} \nabla_a Q(s, a')\big|_{a'=\pi_\theta(s)} \right) \right)^\top \nabla_\theta \pi_\theta(s)$$

$$= \left[ -2\alpha(\pi_\theta(s) - a) + \nabla_a Q(s, a')\big|_{a'=\pi_\theta(s)} \right]^\top \nabla_\theta \pi_\theta(s)$$

The resulting gradients are identical. □

## C  PROOF OF PROPOSITION 2

*Proof.* We are given two conditions that $g(s, a)$ must satisfy:

1. $g(s, a) \propto \nabla_a Q_\phi(s, a)$

2. $\mathbb{E}_{(s,a) \sim \mathcal{D}}[\|g(s, a)\|_2] = \delta$

From the first condition (proportionality), we can write $g(s, a)$ as the gradient scaled by some constant $C$:

$$g(s, a) = C \cdot \nabla_a Q_\phi(s, a)$$

The constant $C$ must be determined. We use the second condition to solve for $C$.

We substitute our expression for $g(s, a)$ into the second condition:

$$\mathbb{E}_{(s,a) \sim \mathcal{D}}[\|g(s, a)\|_2] = \delta$$
$$\mathbb{E}_{(s,a) \sim \mathcal{D}}[\|C \cdot \nabla_a Q_\phi(s, a)\|_2] = \delta$$

Using the properties of norms, we can pull the scalar $C$ out of the L2-norm as its absolute value:

$$\mathbb{E}_{(s,a) \sim \mathcal{D}}[C \cdot \|\nabla_a Q_\phi(s, a)\|_2] = \delta$$

Because $C$ is a constant, we can pull it out of the expectation:

$$C \cdot \mathbb{E}_{(s,a) \sim \mathcal{D}}[\|\nabla_a Q_\phi(s, a)\|_2] = \delta$$

Now, we solve for $C$:

$$C = \frac{\delta}{\mathbb{E}_{(s,a) \sim \mathcal{D}}[\|\nabla_a Q_\phi(s, a)\|_2]}$$

Finally, we substitute this expression for $C$ back into our original equation for $g(s, a)$. To avoid ambiguity with the variables of integration in the expectation, we use $(s', a')$ as dummy variables for the expectation, as shown in the proposition:

$$g(s, a) = \left( \frac{\delta}{\mathbb{E}_{(s',a') \sim \mathcal{D}}[\|\nabla_{a'} Q_\phi(s', a')\|_2]} \right) \cdot \nabla_a Q_\phi(s, a)$$

This completes the proof. $\square$

---

**Algorithm 1** Direct Optimal Action Learning (DOAL)

---

**Require:** Dataset $\mathcal{D}$, policy parameters $\theta$, Q-function parameters $\phi$ [, Value-function parameters $\psi$]
  1: **repeat**
  2:     Update $\phi[, \psi]$ using value loss of Choice
  3:     Update $\theta$ using Equation 16 for BCLoss of Choice
  4: **until** convergence

---

# D    DOAL OBJECTIVES

In summary, the overall algorithm for DOAL is given in Algorithm 1. Similarly, you can replace the line 2 in Algorithm 1 with any Q value learning algorithms. In this section, we explicitly discuss the DOAL objectives. If one replace $a^{\text{target}}$ with dataset action $a$, one recover the baseline model.

## D.1    DIRECTED IMPLICIT Q-LEARNING (DIQL)

DIQL extends the implicit Q-learning framework by introducing a direct optimal action learning approach that explicitly guides policy optimization through value-aware action adjustments. The key innovation lies in replacing the action of the data set $a$ with the optimized target action $a^*$ in the policy loss, thus directly steering the policy towards high-value regions.

$$\mathcal{L}_\pi^{\text{DIQL}}(\theta) = \mathbb{E}_{(s,a)\sim\mathcal{D}} \left[ \exp\left( \alpha(Q_\phi(s,a) - V_\psi(s)) \right) \log \pi_\theta(a^{\text{target}}|s) \right], \tag{18}$$

Notice that the weighting uses the original $Q(s,a)$ as we want to reduce computational overhead.

## D.2    DIRECT IMPLICIT FLOW-Q-LEARNING (DIFQL)

Implicit Flow-Q-Learning (DIFQL) (Park et al., 2025c) builds upon the framework established by Implicit Diffusion Q-learning (Hansen-Estruch et al., 2023). While the updates for the Q-function and value function remain consistent with IQL (see Equation 1), IFQL distinguishes itself by employing a flow-matching objective for policy optimization, as defined by the behavior cloning loss in Equation 3.

The policy's behavior cloning loss in DIFQL is formulated as:

$$\mathcal{L}_\pi^{\text{DIFQL}}(\theta) = \alpha \cdot \mathbb{E}_{a_1\sim\mathcal{N}(0,I),t\sim\mathcal{U}[0,1]} \left[ \|v_\theta(a_t,t) - (a^{\text{target}} - a_1)\|^2 \right], \quad a_t = (1-t)a^{\text{target}} + ta_1 \tag{19}$$

Actions are subsequently generated by sampling from the learned flow model (see Equation 4) and the final action selection is determined by maximizing the learned Q-value function (see Equation 9).

## D.3    EFFICIENT IMPLICIT TRIGFLOW Q-LEARNING (ETRIGFLOW)

We present the Efficient TrigFlow (ETrigFlow) actor objective:

$$L_\pi^{\text{ETrigFlow}}(\theta) = \mathbb{E}_{(s,a_0,t,z)\sim\mathcal{D}} \left[ -Q_\phi(s, f_\theta(a_t,t)) + \alpha \cdot \|f_\theta(a_t,t) - a_0\|_2^2 \right] \tag{20}$$

where $a_t = \cos(t) \cdot a_0 + \sin(t) \cdot z$ and $f_\theta(a_t,t) = \cos(t)a_t - \sin(t) \cdot F_\theta(a_t,t)$,

## D.4    DIRECTED IMPLICIT TRIGFLOW Q-LEARNING (DTRIGFLOW)

For DTrigFlow, we build upon the policy diffusion formulation in Equation 6 and adopt the DOAL framework:

$$\mathcal{L}_\pi^{\text{DTrigFlow}}(\theta) = \mathbb{E}_{a_0\sim p(a_0),z\sim\mathcal{N}(0,I),t\sim\mathcal{U}[0,\pi/2]} \left[ \|f_\theta(a_t,t) - a_0^{\text{target}}\|_2^2 \right] \tag{21}$$

where $a_t = \cos(t)a^{\text{target}} + \sin(t)z$.

### D.5 DIRECTED MULTISTEP FLOW Q-LEARNING (DMFQL)

The DMFQL share the same actor loss as the DIFQL, however it trains the value function through Q learning:

$$\mathcal{L}(\phi) = \mathbb{E}_{(\mathbf{s},\mathbf{a},r,\mathbf{s}',\mathbf{a}')\sim\mathcal{D}} \left[ \| Q_\phi(\mathbf{s},\mathbf{a}) - (r + \gamma\,(Q_{\phi'}(\mathbf{s}',\pi_\theta(s')))) \|_2^2 \right], \tag{22}$$

where $\pi_\theta(\mathbf{s}')$ is sampled from MaxQ sampling with $n_{\text{target\_sample}}$

### D.6 DIRECTED MULTISTEP FLOW REGULARIZED Q LEARNING (DMFReBRAC )

DMFReBRAC introduce an additional regularizor on the Q target.

$$\mathcal{L}(\phi) = \mathbb{E}_{(\mathbf{s},\mathbf{a},r,\mathbf{s}',\mathbf{a}')\sim\mathcal{D}} \left[ \| Q_\phi(\mathbf{s},\mathbf{a}) - \left(r + \gamma\,\left(Q_{\phi'}(\mathbf{s}',\pi_\theta(s')) - \alpha_{\text{critic}} \cdot (\mathbf{a}' - \pi_\theta(\mathbf{s}'))^2\right)\right) \|_2^2 \right], \tag{23}$$

where $\pi_\theta(\mathbf{s}')$ is sampled from MaxQ sampling with $n_{\text{target\_sample}}$ and $\alpha_{\text{critic}}$ controls the regularization strength.

# E  EXPERIMENT RESULTS

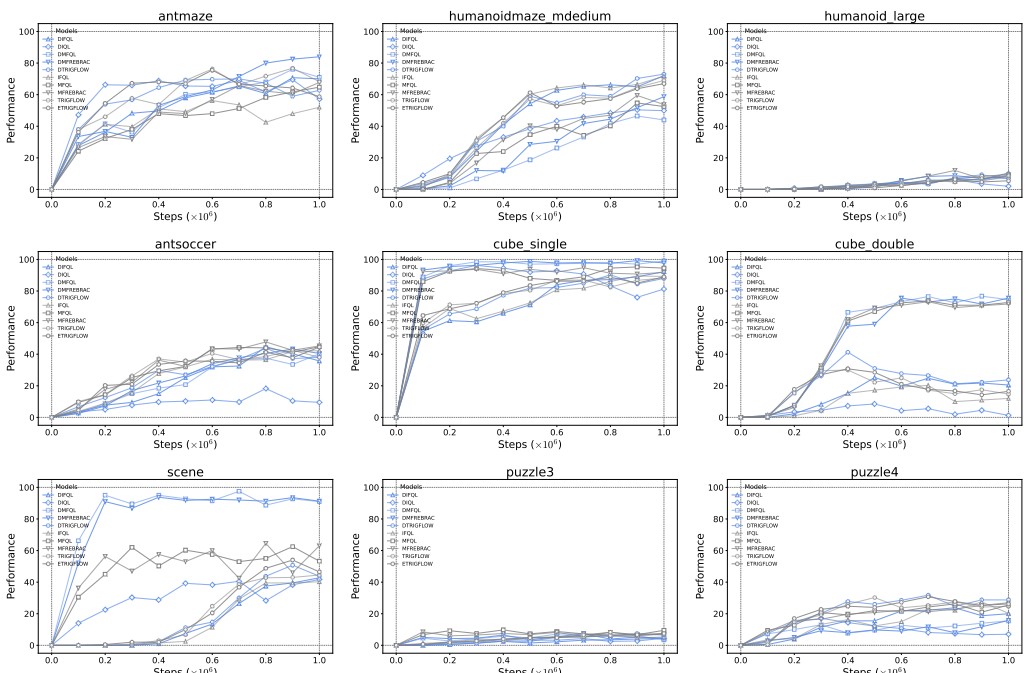

Figure 5: Performance Comparison of Algorithms on the **OGbench** Environment.

The figures 5 illustrate the performance curves of different algorithms across various environments. Specifically, each curve in the generated plots represents the performance trend achieved during the training process of the corresponding algorithm in a specific environment. The vertical axis displays the **normalized return**. This return is the direct evaluation result without applying any aggregation or averaging over multiple runs (e.g., without a moving average filter of size 3). The Antmaze Giant environment is deliberately excluded from this set of figures because the observed performance (normalized return) in these scenes was consistently near zero ($\approx 0$) for all algorithms tested, rendering the resulting plots visually uninformative.

# F ABLATION STUDY

In the following section, we present a detailed discussion on the influence of four critical hyperparameters: the max q sample $n_{\text{sample}}$, the brac coefficient $\alpha$, the $\delta$ in DOAL, and the $n_{\text{target\_sample}}$ for sample next action in DMFQL. Given the number of hyperparameters introduced in this work, and to rigorously demonstrate that our findings are not a consequence of aggressive "tuning", we adopt a strict and transparent two-stage evaluation strategy. Below, we detail the parameter search process and the measures taken to prevent overfitting and selection bias. All hyperparameter choices and initial ablation studies were conducted based on runs using four fixed random seeds: 11, 22, 33, and 44. Crucially, the final, definitive results presented in the main body of this paper are derived from a more robust set of eight independent random seeds: 111, 222, 333, 444, 555, 666, 777, and 888.

With IQL, we first choose $n_{\text{sample}}$ on the Trigflow model, then choose $\delta$ on the DTrigflow model. Then, for all our models with IQL value function $n_{\text{sample}}$ and $\delta$ are shared.

With Q-Learning, we choose $n_{\text{sample}}$ on the MFQL model, and choose $\delta$ on the DMFQL model. On D4RL tasks, we copy the $\alpha_{\text{critic}}$ from ReBRAC and tune it on OGBench.

## F.1 CHOOSING $n_{\text{sample}}$

Table 4 demonstrates that a larger $n_{\text{sample}}$ does not automatically translate into better performance. For instance, across the majority of environments, the optimal performance is achieved at relatively small $n_{\text{sample}}$ values. In contrast, increasing $n_{\text{sample}}$ to 64 or 128 often leads to a measurable decline in the average success rate. Our results demonstrate that MFQL benefits significantly from task-specific optimization of $n_{\text{sample}}$.

Table 4: Comparison of MFQL Performance Across Different $n_{\text{sample}}$ averaging over 4 seeds

| Environment | $n_{\text{sample}}$ | | | | | | | |
|---|---|---|---|---|---|---|---|---|
| | 1 | 2 | 4 | 8 | 16 | 32 | 64 | 128 |
| antmaze-large-navigate-singletask-v0 | 0.050 | 0.655 | **0.695** | 0.560 | 0.330 | 0.265 | 0.140 | 0.125 |
| humanoidmaze-medium-navigate-singletask-v0 | 0.015 | 0.250 | 0.480 | 0.500 | 0.600 | **0.610** | 0.550 | 0.500 |
| humanoidmaze-large-navigate-singletask-v0 | 0.000 | 0.005 | 0.045 | 0.075 | **0.095** | 0.045 | 0.040 | 0.035 |
| antsoccer-arena-navigate-singletask-v0 | 0.005 | 0.165 | 0.355 | **0.500** | 0.460 | 0.400 | 0.295 | 0.285 |
| cube-single-play-singletask-v0 | 0.105 | **0.965** | 0.950 | 0.870 | 0.780 | 0.675 | 0.645 | 0.575 |
| cube-double-play-singletask-v0 | 0.005 | 0.290 | **0.700** | 0.625 | 0.455 | 0.350 | 0.215 | 0.135 |
| scene-play-singletask-v0 | 0.040 | 0.550 | **0.665** | 0.635 | 0.545 | 0.630 | 0.520 | 0.550 |
| puzzle-3x3-play-singletask-v0 | 0.005 | **0.090** | 0.040 | 0.000 | 0.000 | 0.000 | 0.000 | 0.000 |
| puzzle-4x4-play-singletask-v0 | 0.000 | 0.045 | **0.205** | 0.095 | 0.080 | 0.055 | 0.065 | 0.050 |

## F.2 CHOOSING $\delta$

Regarding the $\delta$ hyperparameter, our analysis primarily serves to validate the effectiveness of DOAL. We strategically compared the performance of DMFQL using three distinct $\delta$ values: 0.03, 0.1, and 0.3. As demonstrated in the performance comparison (Figure 6), the choice of $\delta$ has a major impact on the success rate across most environments. While we do not rule out the existence of even better $\delta$ values, our core objective was to show that the active learning mechanism introduced by the $\delta$ parameter is effective and necessary.

It should be noted that $\delta = 0$ recovers the baseline model exactly, but we do not consider this a valid hyperparameter choice.

## F.3 IMPORTANCE OF BEHAVIOR CLONE COEFFICIENT $\alpha$

We kept $\alpha$ from FQL (Park et al., 2025c) in the main experiments for controlling. In Figure 7, on DMFQL, we compare the set $\alpha$ from FQL with $\alpha = 1$. We found that this parameter does not matter. In fact, one should realize $\alpha$ effectively controls the learning rate for actor networks and is no longer in charge of balancing in-distribution vs. Q value. With modern optimizers like ADAM (Kingma & Ba, 2017), such explicit weighting is not needed.

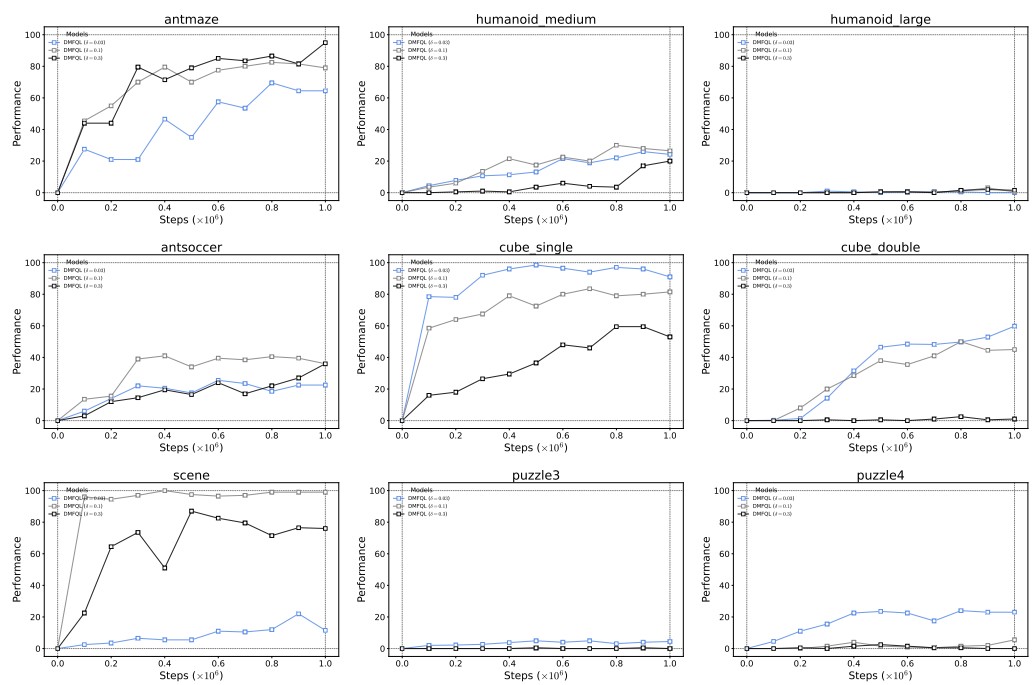

Figure 6: Performance Comparison of **DMFQL** on the **OGbench** Environment with $\delta = 0.03, 0.1, 0.3$ over 4 seeds.

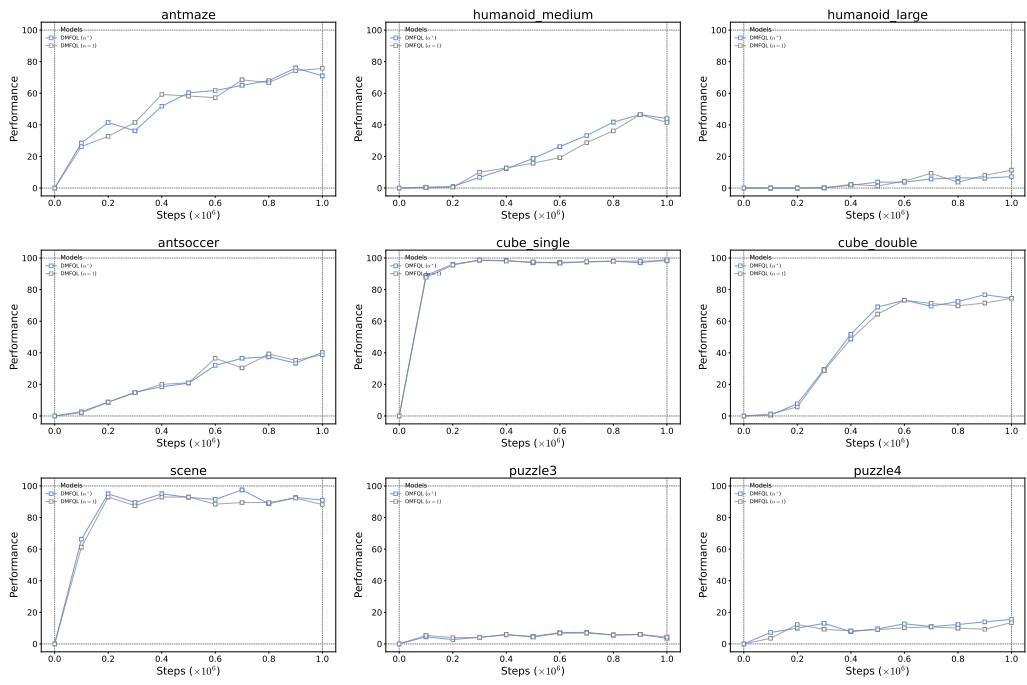

Figure 7: Performance Comparison of **DMFQL** on the **OGbench** Environment with alpha* from **FQL** and alpha = 1 over 4 seeds.

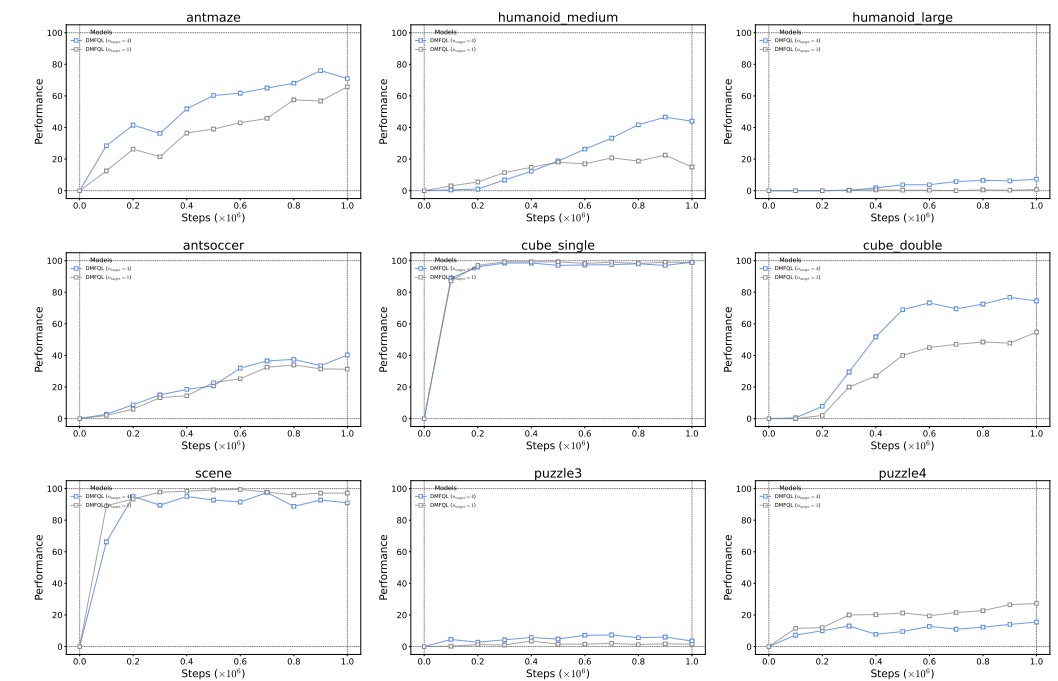

Figure 8: Performance Comparison of **DMFQL** on the **OGbench** Environment with target n sample = 4 and target n sample = 1 over 4 seeds.

### F.4 CHOOSING $n_{\text{target\_sample}}$

The $n_{\text{target\_sample}}$ is necessary for computing the Bellman target value in the DMFQL Q-learning loss:

$$\mathcal{L}(\phi) = \mathbb{E}_{(\mathbf{s},\mathbf{a},r,\mathbf{s}',\mathbf{a}')\sim\mathcal{D}} \left[ \| Q_\phi(\mathbf{s},\mathbf{a}) - (r + \gamma \left( Q_{\phi'}(\mathbf{s}', \pi_\theta(\mathbf{s}')))\right) \|_2^2 \right],$$

where $\pi_\theta(\mathbf{s}')$ is an approximation of $\arg\max_{\mathbf{a}'} Q_{\phi'}(\mathbf{s}', \mathbf{a}')$. To estimate this maximum in a continuous action space, we employ MaxQ sampling by drawing $n_{\text{target\_sample}}$ actions from the policy $\pi_\theta$ at state $\mathbf{s}'$ and taking the maximum Q-value. While MaxQ sampling can run samples in parallel, if the memory is full, increasing $n_{\text{target\_sample}}$ will significantly increase the running time. We fix the $n_{\text{target\_sample}} = 4$ for balancing computational budget and performance. In Figure 8, we observe that $n_{\text{target\_sample}} = 4$ could achieved better results than $n_{\text{target\_sample}} = 1$ overall, but not always. We acknowledge that a more exhaustive hyperparameter search might yield slightly improved results for specific environments; our primary goal was to validate the effectiveness of the DOAL framework, not to achieve the absolute maximum score through extensive tuning.

# G HYPERPARAMETERS

Table 5: DOAL Related Hyperparameters

| Task | IQL-based | | (Regularized) QL-based | | |
|---|---|---|---|---|---|
| | $\delta$ | $n_{\text{sample}}$ | $\delta$ | $n_{\text{sample}}$ | $\alpha_{\text{critic}}$ |
| antmaze-large-navigate | 0.1 | 4 | 0.03 | 4 | 0.01 |
| humanoidmaze-medium-navigate | 0.1 | 32 | 0.1 | 32 | 0.01 |
| humanoidmaze-large-navigate | 0.03 | 8 | 0.03 | 16 | 0.001 |
| antsoccer-arena-navigate | 0.1 | 16 | 0.1 | 16 | 0.1 |
| cube-single-play | 0.03 | 32 | 0.03 | 2 | 0.01 |
| cube-double-play | 0.1 | 16 | 0.03 | 4 | 0.1 |
| scene-play | 0.1 | 32 | 0.1 | 4 | 0.01 |
| puzzle-3x3-play | 0.03 | 4 | 0.03 | 2 | 0.01 |
| puzzle-4x4-play | 0.03 | 64 | 0.03 | 4 | 0.01 |
| pen-human-v1 | 0.001 | 8 | 0.0003 | 4 | 0.5 |
| pen-cloned-v1 | 0.001 | 16 | 0.0003 | 32 | 0.5 |
| pen-expert-v1 | 0.001 | 64 | 0.0003 | 32 | 0.01 |
| door-expert-v1 | 0.003 | 2 | 0.003 | 16 | 0.01 |
| hammer-expert-v1 | 0.003 | 2 | 0.003 | 4 | 0.01 |
| relocate-expert-v1 | 0.003 | 2 | 0.0003 | 4 | 0.01 |

Table 5 summarizes the hyperparameters that we find. However, on D4RL tasks, we find that the variance is high across seeds. The optimal hyperparameters are not reliable. This might be another reason for the lack of improvement of DOAL models.

Table 6: Other Hyperparameters

| Hyperparameter | Value |
|---|---|
| Learning rate | 0.0003 |
| Optimizer | Adam (Kingma & Ba, 2015) |
| Gradient steps | 1000000 (OGBench), 500000 (D4RL) |
| Minibatch size | 256 |
| MLP dimensions | $[512, 512, 512, 512]$ |
| Nonlinearity | GELU (Hendrycks & Gimpel, 2023) |
| Target network smoothing coefficient | 0.005 |
| Discount factor $\gamma$ | 0.99 |
| IQL expectile | 0.9 |
| Flow steps | 10 |
| Flow time sampling distribution | $\text{Unif}([0, 1])$ |

