# OpenReview forum: "Direct Optimal Action Learning"
_ICLR.cc/2026/Conference — Submitted to ICLR 2026_

### Official Review · Reviewer_2U5f · 2025-10-26

**Soundness:** 3
**Presentation:** 2
**Contribution:** 2
**Rating:** 4
**Confidence:** 2

**Summary:**

Direct Optimal Action Learning (DOAL) simplifies offline RL policy extraction by decoupling optimal action computation from policy training. It achieves this by generating a synthetic "optimal" action via Q-function ascent, then uses efficient behavior cloning to train the policy to match it. This framework, which also introduces a more stable trust region hyperparameter, consistently outperforms strong baselines across various policy types and benchmarks.

**Strengths:**

1. Novel Policy Extraction. DOAL elegantly decouples optimal action computation from policy training, sidestepping complex backpropagation through generative models by reframing optimization as simple behavior cloning.
2. Improved Hyperparameter Stability. Reinterpreting the BRAC trade-off coefficient $\alpha$ as a normalized trust region $\delta$, backed by Proposition 2 and the Batch-Normalizing Optimizer, offers a more interpretable and stable hyperparameter. Empirical evidence shows $\delta$ can be shared across algorithms, simplifying tuning.
3. Strong Empirical Results. DOAL achieves consistent and significant performance gains across 15 challenging offline RL tasks, notably improving over meticulously tuned baselines, which enhances the credibility of its effectiveness.

**Weaknesses:**

1. Inconsistency in Theorems and practical implementation: Proposition 1 establishes the gradient equivalence by defining $a^{\text {target }}$ using the Q-gradient evaluated at the policy's output, $\pi_\theta(s)$. However, the paper then argues this is conceptually inconsistent and, in the final DOAL objective (Eq. 16), defines the target using the Q-gradient evaluated at the data action, $a$. While this change is key to decoupling, the original equivalence from Proposition 1 no longer strictly holds, leaving a gap in the theoretical justification for the final objective.
2. The paper introduces the Batch-Normalizing Optimizer and the trust-region parameter $\delta$ as a more stable replacement for the sensitive hyperparameter $\alpha$ from the BRAC objective. Despite this, the final DOAL actor loss in Equation 15 still contains an $\alpha$ parameter, which is described as a controller for the "learning rate of actor" and is copied from a prior work. This is confusing. It is unclear how this $\alpha$ interacts with $\delta$ and why it is still necessary if the optimization trade-off is now managed by the trust region. This ambiguity detracts from the otherwise clean narrative of simplifying hyperparameter tuning.
3. In Table 1, some experimental results with D- still behind the vanilla algorithms. Also, the reviewer suggests that the authors to better present this table such as showing the final averaged performance.

**Questions:**

see weakness

---

> ### Author Response · Authors · 2025-11-24
>
> 1.   Inconsistency: Proposition 1 is presented as an inspiration and tool for comparision between BRAC and DOAL. We are not claiming the DOAL gradient is the same, but rather arguing Direct Optimal Action Learning (DOAL) is reasonable in its’ own right. This is also why, we call out the End2End differentiable optimization is a doctrine.
>
> 2.   Remaining $\alpha$. We kept the $\alpha$ for fair comparison, we provided an ablation where we set $\alpha=1$ in appendix. It turns out that $\alpha$ does not matter.
>
> 3.   Accessible Table: many thanks for the suggestion, we have broke the table a bit more and add summary numbers.
>
>
> Please reconsider the rating based on the new information, and let us know if you have any further questions.

---

### Official Review · Reviewer_5mtx · 2025-10-31

**Soundness:** 2
**Presentation:** 2
**Contribution:** 2
**Rating:** 2
**Confidence:** 3

**Summary:**

The paper proposes Direct Optimal Action Learning (DOAL), a framework for offline RL that reframes behavior-regularized actor–critic training as directly learning an “optimal” action target for each data point and then imitation-learning that target using losses native to the policy family (Gaussian Policies, flow policies and diffusion policies). This avoids costly backpropagation through iterative sampling chains in expressive policies (diffusion/flow), while keeping value guidance from the learned Q-function. The authors also introduce a Batch-Normalizing Optimizer that sets a dataset-level “trust region” (δ) to control how far targets move from behavior actions, aiming to make tuning more interpretable and consistent across policy distributions. Empirically, across 15 tasks from OGBench and D4RL Adroit, DOAL variants match the performance of strong baselines (IQL/FQL/TrigFlow) with shared, environment-level hyperparameters.

**Strengths:**

The paper presents a novel and elegant reformulation of behavior-regularized actor–critic training by directly learning an optimal per-sample action target and then imitating that target with policy-native losses, which removes the need for costly backpropagation through iterative sampling chains in flow and diffusion-based policies. This idea is conceptually clean, improves computational practicality for expressive policy classes, and introduces a trust-region–style batch normalization that provides a more interpretable and stable hyperparameter compared with traditional BRAC scaling. Empirically, across OGBench and D4RL Adroit, the method achieves performance comparable to strong baselines (IQL/FQL/TrigFlow) while using shared environment-level hyperparameters, suggesting that the proposed reformulation maintains effectiveness without extensive tuning

**Weaknesses:**

1. Writing & clarity: The paper has noticeable grammatical issues, inconsistent notation, and several typos, which make the narrative harder to follow; The main results table is dense and difficult to scan, hindering quick cross-method comparisons across policy families and environments; consider adding per-method aggregates (e.g., mean/median across tasks) to improve readability.

2. Shallow experimentation: The evaluation covers only 15 tasks total—6 from D4RL Adroit and 9 from OGBench. Moreover, results are averaged over just 4 seeds (D4RL) and 3 seeds (OGBench), which weakens statistical confidence and robustness of the findings.

3. No compute analysis: A central claimed benefit is reduced computational cost by avoiding backpropagation through iterative diffusion/flow sampling chains, yet the paper provides no wall-clock or memory measurements to substantiate this. Even a simple per-epoch time or training-step latency comparison against BRAC-style training would clarify the practical value.

4. Missing ablations: There are no targeted ablations to disentangle contributions from (i) direct optimal-action targets, (ii) the Batch-Normalizing optimizer; the current analysis discusses hyperparameters but doesn’t isolate causal impact on performance.

**Questions:**

See Weaknesses

---

> ### Author Response · Authors · 2025-11-24
>
> 1.   Writing. Hopefully, the presentation has been improved. If there is anything unclear, please let us know
> 2.   Shallow experimentation: We have increased the number of runs. However, as we are testing our DOAL across policies and Q value functions, it is not feasible for US to run on all available environments. Thousands of computing hours has been spent.
> 3.   No compute analysis: We have included a section in experiment to discuss the neural network forward&backward calss and the actual computing time of DOAL compared to BPTT and FQL.
> 4.   Missing ablations: The Batch-Normalizing optimizer is not meant to improve the final performance but simplifying the hyperparameter search. More concretely, if the batch gradient norm is stable across time (it should be stable across batches as an Monte-Carlo gradient norm estimator for a batch size of 256), we can use C = \frac{\delta}{\text{E}_{(s',a') \sim \mathcal{B}}[\|\nabla{a'} Q_\phi(s', a')\|_2]}  to scale the \nabla_a Q_\phi(s, a). The result will be the same. The drawback of using $C\cdot \nabla_a Q_\phi(s,a)$ is that the range for optimal C will be wider. As for the computational overhead, the normalization is very fast compared to NN calls.
>
>
> Please reconsider the rating based on the new information, and let us know if you have any further questions.

---

### Official Review · Reviewer_GcHL · 2025-10-31

**Soundness:** 3
**Presentation:** 2
**Contribution:** 2
**Rating:** 6
**Confidence:** 4

**Summary:**

The paper presents Direct Optimal Action Learning (DOAL), a framework that replaces end-to-end BRAC-style policy gradients through the value network with a two-step policy extraction:
(1) compute an “optimized” target action per data point via a first-order step on the Q-function;
(2) train the policy to imitate this target with a behavior-native loss (e.g., Gaussian MSE, flow matching, diffusion reconstruction).
This yields distribution-agnostic policy extraction without backpropagating through iterative samplers used by diffusion/flow policies. The authors reinterpret the BRAC trade-off as a trust region and introduce a Batch-Normalizing Optimizer that scales the action step by the batch norm of |\∇ₐQ|, controlled by a single hyperparameter δ. Experiments on OGBench and D4RL Adroit show consistent improvements of DOAL variants (Gaussian, flow, diffusion) over their respective baselines, and analyze Max-Q sampling and the role of candidate count
𝑛
sample
n
sample
	​

.

**Strengths:**

Simple, general recipe for policy extraction. Shows BRAC’s policy gradient can be reframed as target-matching against an action updated by ∇ₐQ, enabling training with any policy family’s native loss (Gaussian/flow/diffusion) and avoiding gradients through multi-step samplers.

Interpretable trust region. The Batch-Normalizing Optimizer controls expected step size via δ, normalizing by batch statistics of |\∇ₐQ|²—cleaner than tuning BRAC’s 𝛼.

Covers expressive policies. Instantiates DOAL for Gaussian, Flow Matching, and Diffusion policies with concrete objectives (e.g., DIQL, DIFQL, DTrigFlow).

**Weaknesses:**

Novelty is mainly a reinterpretation plus a practical recipe.
The equivalence that turns BRAC into target-matching is neat but technically light. Stronger differentiation from value-guided diffusion/flow, energy guidance, and behavior-regularized objectives would clarify what DOAL achieves beyond engineering convenience.

Value-quality dependence and limited uncertainty handling.
DOAL targets rely on ∇𝑎𝑄 at data actions; there’s no integrated uncertainty-aware step control (ensembles, variance-based scaling) for OOD safety, which matters in offline regimes.

Benchmark scope and stability.
While OGBench and Adroit are covered, Adroit volatility and late-training collapse are reported without a full diagnosis, weakening stability claims.

**Questions:**

1. Target evaluation point. You compute ∇𝑎𝑄 at data actions to avoid a mismatch with 𝜋𝜃(𝑠). Have you compared evaluating at 𝜋𝜃(𝑠） (or hybrids) and quantified the difference?

2. δ scheduling. Does adaptive or annealed δ improve late-training stability on Adroit where collapses occur?

3. Value backbones. What happens when pairing DOAL with ReBRAC-style critics—do gains persist?

4. High-D actions. For diffusion/flow in high-D action spaces, does anisotropy in ∇𝑎𝑄 hurt target-matching? Any whitening or per-dimension scaling beyond batch normalization?

5. Max-Q proposals. Beyond sampling from 𝜋𝜃, can proposal refinement (e.g., short Langevin steps guided by Q) reduce noise sensitivity for large 𝑛 samples?

---

> ### Author Response · Authors · 2025-11-24
>
> ***Re-Weakness***
> 1.   Reinterpretation:  This re-interpretation makes the efficient optimization feasible for flow/diffusion models. Hence, we believe this is non-trivial. Also, see overall comment.
>
> 2.  Value dependence:    we have experimented more value functions, and obtained even stronger results.
>
> 3.  D4RL weakness:   we are able to obtain improvements with ReBRAC value function. Yet the final model still lag behind simple ReBRAC. We hypothesis this is due to tanh layer before the action output. By removing tanh ReBRAC with simple Gaussian policy is weaker than our best model.
>
> ***Re-Questions***
> Overall, we consider those questions as suggestions to improve the model. We have obtained improvements by changing the value function.  As we want to make our method simple and avoid overfitting, we try to avoid over-engineering. Nonetheless, we provide some responses.
>
> 3    Action spaces processing: Unfortunately, RL action space is not exactly high-dimensional and not very interpretable. We did not perform any extra processing on the action space. That being said, we do notice a related issue. In D4RL experiment, use tanh to squeeze the action in ReBRAC can explain a lot of its’ advantage. We believe this squeezing allows better modeling of the actions at boundary. This begs the question of the importance of action space geometry. Naively applying tanh^{-t} to convert actions for flow/diffusion modeling could run into NaN. It might be possible that we apply latent diffusion technique to learn a VAE for encoding actions first. This might be worth another paper.
>
> 4    Q-guidance: Testing time Q guided updates are one of the previous methods to improve diffusion model from baselines. We have discussed them in related work section.
>
> Please reconsider the rating based on the new information, and let us know if you have any further questions. For those very constructive questions, if there is one set up you are very curious, we could explore before the final draft.

---

> > ### Comment · Reviewer_GcHL · 2025-11-25
> > **Response to authors**
> >
> > Thank you for the detailed responses. Several of my concerns have been solved, making the results more convincing. Accordingly, I raise the rating confidence but keep the rating score.

---

### Official Review · Reviewer_bqLn · 2025-11-03

**Soundness:** 2
**Presentation:** 3
**Contribution:** 2
**Rating:** 2
**Confidence:** 5

**Summary:**

The paper proposes Direct Optimal Action Learning (DOAL), a framework for offline RL that avoids backpropagating through multi‑step generative policies (diffusion/flow) when using BRAC‑style actor objectives. The key observation is that the BRAC policy gradient is (approximately) equivalent to minimizing the distance between the policy output and a single “optimal action” target derived via a first‑order update from the dataset action. DOAL computes that target directly from \nabla_a Q_\phi(s,a) (evaluated at the dataset action) and then trains the policy to imitate the target using a loss native to the policy family (e.g., flow matching or a diffusion loss), thus decoupling target computation from the policy’s sampling chain. The paper further replaces the sensitive BRAC coefficient \alpha with a Batch‑Normalizing Optimizer that scales the update so that the expected squared step size equals a user‑chosen trust‑region parameter, and presents Algorithm 1 showing integration with IQL for value learning. Experiments on 9 OGBench tasks and 6 D4RL Adroit tasks compare Gaussian, flow, and diffusion policies show mixed results.

**Strengths:**

- Clear, unifying insight and simple objective. This paper makes explicit that BRAC’s policy gradient equals the gradient of a squared‑error to a target action. This clarifies what end‑to‑end training is doing and motivates learning the target directly, which is easy to implement and compatible with any policy class.
- DOAL avoids backprop through iterative sampling chains, which is particularly important for diffusion/flow actors by supervising with native behavior losses. This makes the approach broadly plug‑and‑play with flow/diffusion policies.
- Hyperparameter reinterpretation via a trust region. The Batch‑Normalizing Optimizer replaces a brittle \alpha search with an interpretable \delta that sets an expected squared update magnitude; the denominator’s batch statistic stabilizes scale across tasks.

**Weaknesses:**

- To me, it's unprincipled to replace a’ that which should be sampled from the policy with the dataset action a, the paper does not provide a solid justification for this change (cf. Prop. 1/Eq. 13). In addition, the derivation appears to hinge on an L2 BC loss—how does the argument extend to other losses (e.g., log-likelihood, flow-matching, diffusion objectives)?
- Results are not uniformly stronger. The claim that DOAL “consistently improves performance” over strong baselines is only partially supported. For example, DTrigFlow is dramatically worse on cube‑single‑play, and DIQL is broadly weaker than IQL on many tasks.
- Efficiency claims lack measurement. A central motivation is avoiding BPTT through iterative samplers, but there are no wall‑clock time, memory, or gradient‑call counts comparing DOAL to TrigFlowQL/FQL under matched hardware.
- The observation of Max-Q sampling does not constitute a novel contribution. Prior work (e.g., Q-chunking) already formalizes that the candidate count N in Max-Q sampling implicitly sets the level of KL regularization between the induced policy and the proposal/sampling distribution. From this perspective, increasing N weakens the effective regularization and can degrade performance, so it is unsurprising that “the bigger N, the better” does not hold. Please cite these results..
- Scope of value learners. Most experiments couple DOAL with IQL for a controlled study, but since BRAC‑style methods remain strong (e.g., ReBRAC in Table 1). Is this policy extraction approach still improve on other value learners?

**Questions:**

see Weaknesses.

---

> ### Author Response · Authors · 2025-11-24
>
> We sincerely thank the reviewer for their knowledgeable and constructive feedback.
>
> **L2 BC loss limitation:** We are not claiming DOAL is same as BRAC, see overall feedback. In terms of analyzing with other BC losses, we note that L2 loss is equivalent to Gaussian log-likelihood. As for models requiring BPTT, their trouble with BRAC is precisely it is hard to understand the nature of what they are doing. While in DOAL, we are doing direct optimal action learning. We supplemented with an BPTT experiment, and indeed found that it is less stable.
>
> **Weak Results.** See overall response. We believe the results are sufficiently strong now. The cube-single drop is actually a typo.
>
> **Efficiency.** We have included a new section on computing time. Intuitively, DOAL only cost one extra forward and backward calls on Q value net, compared to baselines. In terms of actual run time.
>
> **The observation of Max-Q sampling:** Many thanks for the referred paper, we believe this is indeed relevant and Q-chunking has searched over number of samples. However, we still believe our insight is complementary. The regularization story only tells what happens when number of samples is small, We are arguing that large number of samples will destroy model output even if your Q estimator is unbiased. Meanwhile, we do not believe that regularization on one side implies performance degragation on the other side. In fact, if one were to construct some weighted resampling techniques （instead of maxQ, but still enjoys regularization with small number of samples）, increasing the number of samples to the extreme side might not be bad as it could keep the regularization effect.
>
> Moreover, Q-chunking is on NIPS this year, not exactly a prior work (it might be on arxiv at the time of our submission, but with the title unrelated to our paper and maxQ). Prior works did not perform search on this parameter to our best knowledge, and contain some misbeliefs. we added more discussion on the related work section.
>
> **Scope of value learners.** Many thanks for the suggestion, in fact, we can see more reliable gains with ReBRAC.
>
>
> Please reconsider the rating based on the new information, and let us know if you have any further questions.

---

### Author Response · Authors · 2025-11-24
**Main Experimental Updates**

| Task                                   | IQL(tanh) *   | IQL(Gauss)   | DIQL   | IFQL*  | IFQL   | DIFQL  | TrigFlow | DTrigFlow | ETrigFlow |
|------------------|--------|--------|--------|--------|--------|--------|----------|-----------|-----------|
| antmaze-large-navigate              | –      | 48 ± 9    | 63 ± 10| 24 ± 17| 48 ± 24| 67 ± 6 | **72 ± 6**| 63 ± 24   | 63 ± 21   |
| humanoidmaze-medium-navigate   | –         | 32 ± 7      | 55 ± 8 | **69 ± 19**| 68 ± 3 | 68 ± 4 | 64 ± 4   | 67 ± 4    | 63 ± 4    |
| humanoidmaze-large-navigate      | –       | 3 ± 1        | **10 ± 3**| 6 ± 2  | 6 ± 3  | 8 ± 3  | 7 ± 2    | 8 ± 4     | 6 ± 3     |
| antsoccer-arena-navigate          | –      | 3 ± 2        | 13 ± 3 | 16 ± 9 | 40 ± 5 | 40 ± 6 | 40 ± 8   | **41 ± 4**| **41 ± 9**|
| cube-single-play                   | –     | 85 ± 8     | 80 ± 4 | 73 ± 3 | 88 ± 4 | **90 ± 3**| 86 ± 4   | 88 ± 2    | 88 ± 4    |
| cube-double-play                 | –       | 1 ± 1       | 3 ± 2  | 9 ± 5  | 11 ± 3 | 21 ± 4 | 16 ± 4   | **22 ± 3**| 16 ± 3    |
| scene-play                         | –     | 12 ± 3      | 37 ± 10| 0 ± 0  | 40 ± 23| 40 ± 23| 43 ± 16  | 46 ± 15   | **50 ± 12**|
| puzzle-3x3-play                  | –       | 2 ± 1      | 5 ± 1  | 0 ± 0  | 5 ± 1  | 5 ± 2  | 7 ± 2    | 7 ± 3     | **8 ± 2** |
| puzzle-4x4-play                 | –        | 5 ± 2       | 10 ± 2 | 21 ± 11| 23 ± 7 | 21 ± 5 | 26 ± 5   | **27 ± 6**| 26 ± 4    |
| **Total**                        | –       | 191        | 276    | 218    | 329    | 359    | 361      | **368**   | 359       |
| **Flow Policies**                      |        |        |        |        |        |        |          |           |           |
| pen-human-v1                           | 78     | 54 ± 6 | 43 ± 8 | 71 ± 12| **81 ± 8**| 68 ± 8 | 71 ± 11  | 69 ± 13   | 72 ± 12   |
| pen-cloned-v1                          | **83** | 66 ± 7 | 56 ± 9 | 80 ± 11| 73 ± 7 | 74 ± 7 | 65 ± 7   | 67 ± 8    | 67 ± 9    |
| pen-expert-v1                          | 128    | 131 ± 8| 132 ± 4| **139 ± 5**| 134 ± 4| 138 ± 4| 135 ± 8  | 133 ± 7   | 134 ± 8   |
| door-expert-v1                         | **107**| 104 ± 2| 104 ± 2| 104 ± 2| 104 ± 1| 104 ± 1| 104 ± 1  | 104 ± 1   | 104 ± 1   |
| hammer-expert-v1                       | **129**| 68 ± 47| 76 ± 46| 117 ± 9| 96 ± 8 | 98 ± 12| 103 ± 8  | 98 ± 11   | 100 ± 10  |
| relocate-expert-v1                     | 106    | 97 ± 10| 101 ± 5| 104 ± 3| 104 ± 3| 102 ± 8| 106 ± 2  | **107 ± 2**| 106 ± 2   |
| **Total**                              | **631**| 520    | 518    | 615    | 592    | 584    | 584      | 577       | 583       |

| Task | FQL* | MFQL | DMFQL | MFReBRAC | DMFReBRAC |
|------|------|------|-------|----------|-----------|
| antmaze-large-navigate | 80 ± 8 | 62 ± 11 | 72 ± 8 | 65 ± 13 | **83 ± 7** |
| humanoidmaze-medium-navigate | 19 ± 12 | 49 ± 9 | 44 ± 13 | 53 ± 14 | 52 ± 7 |
| humanoidmaze-large-navigate | 7 ± 6 | 8 ± 3 | 7 ± 3 | **9 ± 4** | 8 ± 2 |
| antsoccer-arena-navigate | 39 ± 6 | 43 ± 6 | 37 ± 5 | **45 ± 5** | 41 ± 6 |
| cube-single-play | 97 ± 2 | 95 ± 1 | 98 ± 1 | 91 ± 5 | **99 ± 1** |
| cube-double-play | 36 ± 6 | 72 ± 4 | **75 ± 6** | 74 ± 4 | **75 ± 3** |
| scene-play | 76 ± 9 | 57 ± 20 | 90 ± 10 | 57 ± 12 | **92 ± 6** |
| puzzle-3x3-play | **16 ± 5** | 7 ± 3 | 6 ± 2 | 7 ± 2 | 5 ± 2 |
| puzzle-4x4-play | 11 ± 3 | 24 ± 3 | 14 ± 4 | **25 ± 5** | 12 ± 3 |
| **Total** | **381** | **418** | **443** | **425** | **466** |
| pen-human-v1 | 53 ± 6 | 75 ± 9 | 72 ± 8 | 64 ± 9 | 74 ± 8 |
| pen-cloned-v1 | 74 ± 11 | 75 ± 9 | 80 ± 5 | 71 ± 12 | 75 ± 10 |
| pen-expert-v1 | 142 ± 6 | 138 ± 4 | 130 ± 8 | 140 ± 9 | 143 ± 4 |
| door-expert-v1 | 104 ± 1 | 104 ± 2 | 104 ± 1 | 105 ± 8 | 105 ± 1 |
| hammer-expert-v1 | 125 ± 3 | 126 ± 3 | 124 ± 5 | 126 ± 3 | 126 ± 1 |
| relocate-expert-v1 | 107 ± 1 | 106 ± 4 | 104 ± 4 | 106 ± 1 | 107 ± 2 |
| **Total** | **605** | **623** | **614** | **614** | **630** |

In the first draft, we tested on IQL with limited runs for both hyperparameter search and actual experiments. In the updated version, we increase the runs for hyperparameter selection to 4 and actual runs to 8. In addition, we experimented with  Q learning without and without regularization. MFQL   are multi-step flow models  with standard Q learning. MFReBRAC  are multi-step flow models trained with  ReBRAC objective (regularized Q learning). DMFQL and DMFReBRAC  are their DOAL improvement. (See paper for the ReBRAC results.)

We note that the DOAL model with $\delta=0$ actually recovers the baseline models. However, we do not consider $\delta=0$ as a hyper-parameter candidate. Such non-improvement might be due to the limited extrapolation ability of learned Q value function. A better value estimation is of vital importance to offline RL, however, it is beyond the scope of this paper. Lastly, ReBRAC has advantage from tanh activation. How to effectively build the tanh activation after flow/diffusion models is also interesting.

---

> ### Comment · Area_Chair_N3MH · 2025-11-24
> **Author-Reviewer Discussion**
>
> Dear reviewers,
>
> Please review the authors' response and adjust your rating accordingly. If you have any further questions, please discuss with the authors further.
>
> AC

---

### Author Response · Authors · 2025-12-03
**Summary for Area Chair**

Given the ICLR circumstance, we provide a quick summary for AC.

✍️ Summary

---

This paper propose to learn diffusion/flow policies via **direct optimal action learning** (DOAL). Instead of end2end learning with multi-steps sampling, DOAL propose to compute the "optimal action"  via Taylor expansion regarding Q value around data point, then use efficient distribution matching loss to model the "optimal action". Consequently, a trust region size around Q value is proposed as an interpretable and stable hyper-parameter.

---

Overall, the main reason for the low scores in the initial review is the "weak" experiments. We aimed at fair comparisons between DOAL models and strong baselines under well-controlled IQL value function. The absolute performance of our models were not very high. We now have new experiments with Q learning/ regularized Q learning based flow models. Those models achieved very strong absolute performance, **suppressing the previous best model** Flow Q-Learning.

---
✍️  Strength Summary

---
| Strength | Reviewers |
| :--- | :--- |
| Simple |  bqLn GcHL 5mtx 2U5f |
| Elegent | 5mtx 2U5f |
|Novel | 5mtx 2U5f |
|General | bqLn GcHL  |
|Trust region is interpretable and stable | bqLn GcHL 5mtx 2U5f |
|Strong Relative Performance| 5mtx 2U5f |
|   |   |

All reviewers agree that our method is simple and more interpretable. Some emphasize the elegency/novelty and others point out the generality of our method.  Notice that those comments are based on the initial draft, where we only tested over the IQL value function.

---
✍️  Weakness Summary
---
| Weakness/Question| Reviewers | Our Response and Solution |
| :--- | :--- |  :--- |
| **Insufficient Results**  We need more seeds and performance is not strong enough| bqLn, 5mtx, gchl, 2U5f  |  The results are now **sufficiently strong**. We have **increased the number of runs** for the IQL models. We train flow models with (regularized) Q-learning value function. DOAL models out perform baseline models. || **Insufficient Results**  We need more seeds and performance is not strong enough| bqLn, 5mtx, gchl, 2U5f  |  The results are now **sufficiently strong**. We have **increased the number of runs** for the IQL models. We train flow models with (regularized) Q-learning value function. DOAL models out perform baseline models. |
| **$\alpha$** the behaviour loss scalling factor is kept|  2U5f  | We kept using this hyperparameter from FQL for consistency. We provide ablation study, and find it does not matter much. Setting it to 1 slightly weakened the performance. |
| **Efficiency Analysis** Discussion of Computational Efficiency is Missing|  bqLn, 5mtx |Intuitively, DOAL only costs **one extra forward and backward call** on the Q-value network compared to baselines. We thought it is obvious that DOAL is fast. We have included a new section in the experiment to discuss the computational complexity and actual run time, showing its efficiency compared to BPTT and FQL. |
| **Novelty of Number of Samples in Max-Q Sampling** | bqLn | We thank the reviewer for the Q-chunking reference and have added more discussion. However, the reference is a concurrent work from NIPS this year. Furthermore, our insight is complementary. Regularization view in Q-chuncking only tells what happens when the number of samples is small, it does not necessarily imply large samples size is bad; **we argue large samples can destroy model output** even with an unbiased Q-estimator. Prior work did not perform a search on this parameter or address this concern. We added discussion on the related work section |
| **Effectiveness of Batch-Normalizer** Batch-Normalizer is not tested against scaling Q gradient | 5mtx, 2U5f  | The optimizer is meant to **simplify hyperparameter search**, not improve final performance. We demonstrated experimentally that batch mean of the gradient norm is stable during training, therefore all batch-normalizer has an equivalent direct scalling version. The difference is that the range of later is wider across tasks.|
|   |   |

✍️ Additional Ablation

---

In addition to addressing reviewers questions, we have added other ablation studies on hyperparameters, including number of samples for MaxQ sampling, DOAL $\delta$, number of samples in MaxQ sampling for Q learning target. All the experimental results have been updated in the draft. We present the new main result in another comment.

---

✍️ Conclusion

---
We have significantly strengthened our empirical results and clarified confusing points. GcHL (rating 6 confidence 4) is the only reviewer who has responded by agreeing that concerns have been addressed.

We also identified the source of advantage of ReBRAC in D4RL tasks: tanh activation function. Meanwhile, we acknowledge that gradient based policy extraction might depends on the quality of data and Q value function.

---

### Meta-Review · Area_Chair_rXCB · 2026-01-03

**Summary:**

The reviewers raised concerns from many perspecitves, including the motivation (Reviewer bqLn), the gap between theory and practical implementation (Reviewer 2U5f, and it is also a part of motivation), and the experiments (all reviewers). While the authors added more experiments with more runs, which addressed the reviewers' concerns about the experiments (although partially), other concerns are not sufficiently addressed. Moreover, the manuscript still requires improvements in writing and clarity. Based on these points, I recommend rejection.

**Reviewer Concerns:**

Reviewer bqLn questioned the fundamental justification for replacing policy-sampled actions with dataset actions and noted that the method's performance gains are not consistent across all strong baselines. The reviewer also identified a lack of empirical efficiency measurements and argues that the observation of Max-Q sampling does not constitute a novel contribution. do not constitute a novel contribution over prior work. I have the same feeling with this reviewer on the point that replacing policy-sampled actions with dataset actions needs further discussion, while I do not find the authors' rebuttal sufficiently address this.

Reviewer GcHL questioned the novelty of the paper. The reviewer also highlighted concerns regarding no uncertainty handling for OOD safety and limited benchmark scope and stability. Several of concerns have been addressed, while I think the discussion related to OOD is needed, as it matters in offline RL.

Reviewer 5mtx noted that writing and clarity need to be improved, and the experiments and results are insufficient (e.g., no wall-clock time reported and no ablation study). While additional experimental results have been provided, the conern of the shallow experimentation remains if I were the reviewer, especially w.r.t. the limited number of tasks.

Reviewer 2U5f pointed out that there is inconsistency in Theorems and practical implementation. The reviewer also found that the parameter $\alpha$ was still used in the paper, which contradicts the previous assertion.

All reviewers pointed out that the experimental results are not satisfied, including the performance, number of runs, and scope.

**Reviewer Scores:**

I do not think they will change their score.

---

### Decision · Program_Chairs · 2026-01-26

Reject